

# Implementation of Real-Time Source Apportionment Approaches Using the ACSM-Xact-Aethalometer (AXA) Set-Up with SoFi RT: The Athens Case Study

Manousos I. Manousakas[1,2], Olga Zografou[2], Francesco Canonaco[3], Evangelia Diapouli[2], Stefanos Papagiannis[2], Maria Gini[2], Vasiliki Vasilatou[2], Anna Tobler[3], Stergios Vratolis[2], Jay G. Slowik[1], Kaspar R. Daellenbach[1], André S. H. Prevot[1*], Konstantinos Eleftheriadis[2*]

[1]PSI Center for Energy and Environmental Sciences, Villigen, 5232, Switzerland
[2]Environmental Radioactivity & Aerosol Tech. for Atmospheric & Climate Impacts, INRaSTES, National Centre of Scientific Research "Demokritos", Ag. Paraskevi, 15310, Greece
[3]Datalystica Ltd., Parkstrasse 1, 5234 Villigen, Switzerland

*Correspondence to*: André S. H. Prevot (andre.prevot@psi.ch) and Konstantinos Eleftheriadis (elfther@ipta.demokritos.gr)

**Abstract.** Air pollution, particularly from particulate matter (PM), poses serious public health and environmental risks, especially in urban areas. To address this, accurate source apportionment (SA) of PM is essential for effective air quality management. Traditional SA approaches often rely on offline data collection, limiting timely responses to pollution events. SA applied on data from online techniques, especially with high temporal resolution is advantageous over offline techniques, enabling the study of the diurnal variability of emission sources and also the study of specific events. Recent technological advancements now enable real-time SA, allowing continuous, detailed analysis of pollution sources. This study presents the first application of the ACSM-Xact-Aethalometer (AXA) setup combined with SoFi RT software for real-time source apportionment (RT-SA) of PM in Athens, Greece. The AXA setup integrates chemical, elemental, and black carbon data streams, covering a broad spectrum of PM components and capturing a comprehensive representation of PM sources in an urban environment. The results demonstrate that traffic-related emissions are the largest contributors to PM, with significant contributions from secondary species such as sulfate, nitrate, ammonium, and secondary organic aerosols, which together accounted for approximately 57% of the PM mass. Primary sources such as biomass burning and cooking contributed around 10% each, with natural sources like dust and sea salt comprising the remainder. The SoFi RT software is employed for continuous SA, offering automated analysis of PM sources in near real-time (minutes after the measurements). Our findings demonstrate that this setup effectively identifies major pollution sources. This work underscores the AXA system's potential for advancing urban air quality monitoring and informs targeted interventions to reduce PM pollution.

## 1 Introduction

Air pollution, particularly the presence of particulate matter (PM), continues to be a significant concern in urban environments due to its adverse effects on public health and the environment (Cheung et al., 2024; Glojek et al., 2024; Katsouyanni et al., 1995; Morawska & Zhang, 2002). To effectively manage and mitigate air quality issues, it is crucial to understand the specific



sources contributing to PM levels. Source apportionment (SA), the process of identifying and quantifying these sources (Hopke, 2016), is a critical tool in air quality management. Traditional methods of source apportionment, however, often rely

on offline analysis, which can introduce delays in data collection and limit the ability to respond promptly to pollution events. This has led to the growing need for real-time, continuous source apportionment techniques that allow for faster, more detailed insight into air pollution sources (Chen, Canonaco, Slowik, et al., 2022b).

In recent years, the implementation of real-time SA techniques has become possible due to significant advancements in measurement technology and data processing capabilities. The development of high-resolution, real-time monitors, combined

with powerful computational tools, enables the continuous collection and analysis of air quality data. These systems can now deliver near-instantaneous information about the composition and sources of particulate matter, allowing for more dynamic air quality management (Ng et al., 2011, Drinovec et al., 2015; Fröhlich et al., 2013; Furger et al., 2020). This real-time capability represents a major shift in how air pollution is monitored and managed, enabling more effective interventions and policy decisions aimed at reducing pollution exposure in urban environments.

The available online instruments offer the capability to measure various PM components. However, since no single instrument can characterize all components, it is crucial to use a combination of instruments that collectively provide comprehensive information, capturing most of the PM mass. Additionally, these instruments must produce data with the same time resolution to ensure compatibility for use in source apportionment approaches. One instrumental set-up that can cover the entire range of components is the Aerosol Chemical Speciation Monitor (ACSM), Xact multi-metal monitor, Aethalometer (AXA) set-up.

The ACSM measures the chemical composition of non-refractory submicron particles ($PM_1$) in real-time, including key species such as sulfate ($SO_4^{2-}$), nitrate ($NO_3^-$), ammonium ($NH_4^+$), and organic aerosols (Ng, Herndon, et al., 2011). The ACSM is especially valuable for identifying secondary-like organic aerosol (SOA) formation, traffic, cooking and biomass burning emission, as these sources are often rich in organic particulate components (Chen, Canonaco, Tobler, et al., 2022b). The Xact multi-metal monitor offers real-time measurements of elements in ambient PM. Trace metals are critical markers for a variety

of pollution sources, particularly those related to industrial activities, traffic (e.g., brake and tyre wear), and combustion processes. By continuously monitoring the elemental composition of PM, the Xact instrument helps to pinpoint both natural sources (e.g., dust) and anthropogenic activities (e.g., industrial emissions), which are crucial for a complete understanding of the PM burden in urban areas (M. Manousakas et al., 2021, 2022). Complementing the ACSM and Xact, the Aethalometer measures black carbon (BC) concentrations in real-time. Black carbon is a primary component of PM that originates from the

incomplete combustion of liquid and solid fuels, making it a key indicator of traffic-related emissions (e.g., diesel exhaust) and residential wood burning, respectively. The Aethalometer's capability to differentiate between these sources by analyzing the wavelength-dependence of light absorption provides further specificity in source apportionment (Zotter et al., 2017). Given the strong association between black carbon and both health risks and climate impacts, its measurement is crucial for both public health and environmental policy. The AXA setup represents a significant advancement in real-time air quality

monitoring, offering a comprehensive dataset that captures a wide range of PM characteristics. Each instrument in the setup



plays a distinct role in measuring different aspects of PM composition, and together, they provide a nearly complete picture of the particulate matter mass.

Even though advances in instrumentation have made near real-time source apportionment (RT-SA) approaches possible, efforts in this area remain quite limited. Chen and co-authors demonstrated the application of an RT-SA technique for organic
aerosols in three European cities analyzing ACSM data with an earlier version of SoFi RT (Chen, Canonaco, Slowik, et al., 2022a). The results indicate that the RT-SA can provide very comparable results to best case source apportionment approaches, if the RT-SA is set up properly. This study, even though it implemented state of the art optimized SA approaches, did not refer to the total PM mass, but only on the organic fraction. In a study conducted in Shenzhen, China a combination of instruments that provided information about most of PM mass was utilized (Yao et al., 2024). In this study not all species that are provided
from the instruments were used in the source apportionment analysis (e.g. only m/z 44 was used from Q-ACSM, and six elements from the Xact). In another study that took place in Delhi, India, an RT-SA methodology that reports the results online has been set up (Prakash et al., 2021). In this study, data were collected from an Xact, an Aethalometer, a total carbon analyzer, and low-cost sensors. Due to the nature of the input data, the apportionment focused primarily on the speciation of elements, with no information provided about secondary species.

Source apportionment analysis is influenced by several critical factors, including uncertainties in the data, the number of variables involved, and internal correlations between the variables, particularly when integrating data from multiple instruments. Each of these factors affects the accuracy and reliability of the source attribution process.

When combining data from different instruments such as an ACSM, an Aethalometer, and an Xact, the precision of each instrument in detecting specific pollutants varies. These uncertainties propagate through the apportionment model and can
reduce the confidence in the derived source contributions. The number of variables used in source apportionment, the temporal variation of the fingerprints of the sources, the degree and frequency of transient sources, as well as the internal correlation of the variables also play a significant role. The ACSM measures mass spectral data that includes multiple fragments from the same parent molecules and hence are internally correlated with each other. Combining a large number of internally correlated variables (ACSM), to a much fewer number of independent variables (Xact, Aethalometer), can lead to SA results that are not
equally based on all instruments. In the literature, there are studies that report combining all variables from the ACSM and Xact in a meaningful way to obtain comprehensive source apportionment results; however, none of these studies has applied real-time source apportionment techniques (Belis et al., 2019; Yao et al., 2024), or they are not using all available variables (Zhang et al., 2023).

Athens, Greece, is an ideal case study for implementing this advanced monitoring approach. The city experiences a complex
mixture of pollution sources, including local traffic, industrial activities, residential heating, and regional biomass burning, all of which contribute to its air quality challenges (Diapouli et al., 2016; M. Manousakas et al., 2021). Additionally, these sources vary significantly over time due to weather conditions, seasonal changes, and daily traffic patterns. The real-time source apportionment provided by the AXA setup offers the potential to better characterize these sources and their fluctuations, enabling more effective mitigation strategies.



To harness the full potential of the data generated by these instruments, the SoFi RT (Source Finder Real-Time) software is employed for continuous source apportionment. SoFi RT applies advanced statistical methods to separate and quantify the different sources contributing to the PM burden, providing real-time insights into pollution events and their temporal variability. This model is designed to automatically collect, treat, and use the data for source apportionment analysis in multiple flexible days, which, to the best of our knowledge, makes it the only commercially available model with such capabilities.

Similar capabilities are demonstrated in a software developed under the framework of Clean Air China project that is currently available for the participants of this project. This software offers RT capabilities but with very limited option compared to SoFi RT (rolling window, BS and PR analysis, criteria based selection, averaging, etc).

In this paper, we present the first application of the AXA setup combined with SoFi RT in Athens. We demonstrate the system's ability to deliver real-time, continuous source apportionment by integrating chemical, elemental, and black carbon data

streams. This study highlights the effectiveness of this novel approach in capturing the majority of the PM mass, providing a comprehensive understanding of the primary pollution sources in Athens. Our findings offer important insights for improving air quality management and developing targeted interventions to reduce pollution levels.

## 2 Methodology

### 2.1 Sampling

The Demokritos station (DEM), positioned at 270 meters above sea level (37.995° N, 23.816° E), is a vital hub for atmospheric monitoring and research. It is integrated into major research initiatives, such as the Global Atmosphere Watch (GAW) program, Aerosol, Clouds, and Trace Gases Research Infrastructure (ACTRIS), and PANhellenic infrastructure for Atmospheric Composition and Climate Change (PANACEA). The station is located on the National Centre for Scientific Research (NCSR) "Demokritos" campus, within a vegetated area at the foot of Mount Hymettus, approximately 8 kilometers northeast of Athens'

city center. Its location provides a unique vantage point for capturing suburban aerosol dynamics influenced by urban pollution under prevailing westerly winds and regional contributions during specific atmospheric conditions.

From March 1 to March 31, 2024, measurements of non-refractory PM1 components, i.e., organic matter, sulfate ($SO_4^{2-}$), nitrate ($NO_3^-$), ammonium ($NH_4^+$), and chloride ($Cl^-$), were conducted using a time-of-flight aerosol chemical speciation monitor (ToF-ACSM) developed by Aerodyne Research Inc. (Billerica, MA, USA). This instrument operated with a time

resolution of 10 minutes, and data were subsequently averaged to 30-minute intervals for analysis. Detailed operational parameters and calibration procedures for the ToF-ACSM are provided in (Zografou et al., 2022).

Equivalent black carbon (eBC) concentrations were monitored during the same period using an Aethalometer AE33 (Magee Scientific Corp., Berkeley, CA, USA), which employs the DualSpot Technology to compensate for the filter matrix and filter loading effects in real time (Drinovec et al., 2015), while the multiple scattering is compensated by the H factor provided by

ACTRIS. In this study, eBC concentrations were reported at a wavelength of 880 nm, with a mass absorption cross-section



(MAC) value of 4.6 m² g⁻¹, as recommended by (Kalogridis et al., 2017). Contributions of solid (eBCsf) and liquid (eBClf) fuel sources to eBC were quantified using the Aethalometer model by Sandradewi et al. 2008.

Hourly concentrations of 37 elements (Al, Si, P, S, Cl, K, Ca, Sc, Ti, V, Cr, Mn, Fe, Co, Ni, Cu, Zn, Ga, Ge, As, Se, Br, Rb, Sr, Y, Zr, Nb, Cd, In, Sn, Sb, I, Ba, Hg, Tl, Pb, and Bi) in PM2.5 were measured from March 4 to March 26, 2024, using an

Xact 625i ambient metals monitor. Air was drawn through a filter tape at a flow rate of 16.7 liters per minute over a one-hour sampling interval. The filter tape was then analyzed in an x-ray chamber with a rhodium anode (50 kV, 50 W) under three sequential energy settings optimized to target specific element groups. Calibration and performance were verified using Micromatter standards, ensuring high accuracy in elemental measurements. The elements included in the analysis were Si, S, Cl, K, Ca, Ti, V, Cr, Mn, Fe, Ni, Cu, Zn, As, Br, Sr and Pb.

**2.2 Source apportionment**

**2.2.1 Optimized source apportionment analysis (OP SA)**

The implementation of source apportionment can be distinguished into two different main processes: the Optimized Source Apportionment analysis (OP SA), and the Real time Source Apportionment analysis (RT SA). The flow chart of these approaches is presented in Figure 1. The first step in the source apportionment approach is to assess the situation in the area

by conducting the OP SA. This involves utilizing all available tools and adhering to a classical offline analysis protocol, where data are analyzed at the conclusion of the campaign. OP SA serves two purposes: it acts as a verification method for RT-SA and establishes the baseline conditions in the area; the data generated can be used to set up RT-SA during its initial stages.

In this study, an initial OP SA was conducted using all available data, analyzed separately for the Xact (elemental content), AE33 (BC data), and ACSM (organic content) datasets. Regarding the implementation of the RT, the model's performance

was tested using optimal initial parameters and under less optimized, generic conditions. For the optimized operation of the RT, the number and types of sources (source profiles) identified by the OP SA were used as initial parameters to configure the RT SA run. In both cases, the results of the RT SA were subsequently compared with those of the OP SA as a method of evaluation and verification. The RT was implemented using the same data as the OP SA, following the conclusion of the campaign on simulating conditions. During this process, the raw files from the instruments were input into the RT model

without any pretreatment, replicating normal operational conditions. This approach ensures a stable environment for evaluating the model's performance, free from potential instrumental failures that could affect the results. Detailed descriptions of the implementation of each step are provided in the following sections.




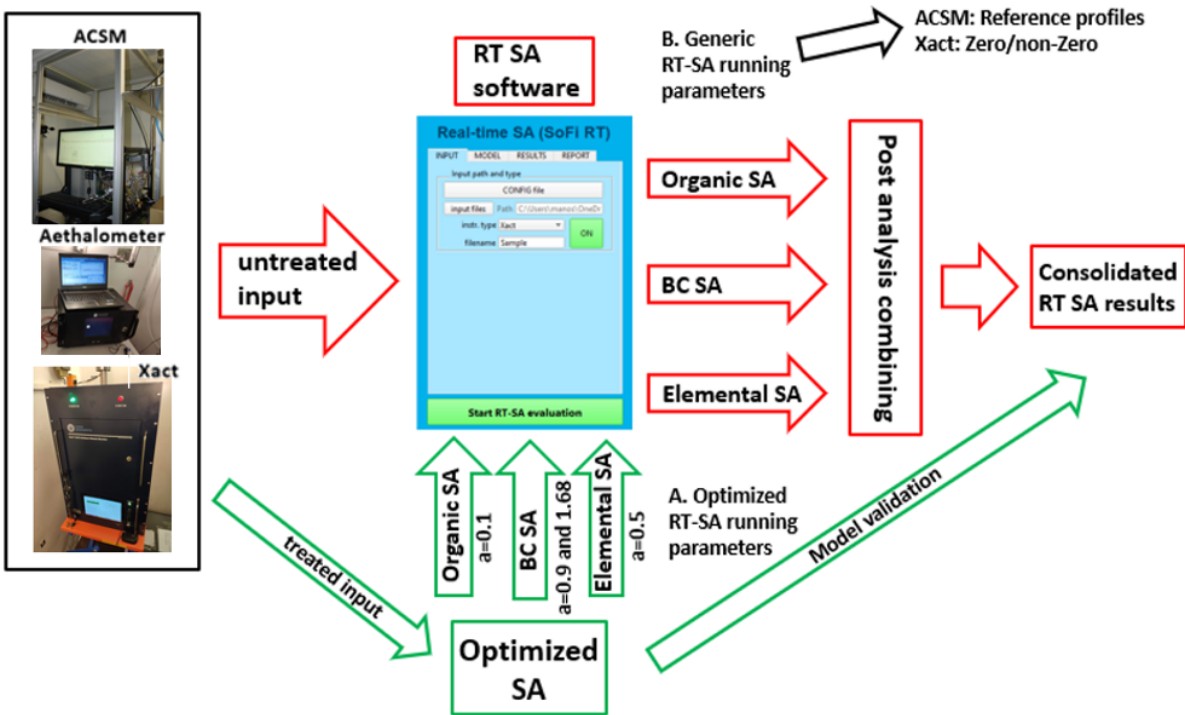

**Figure 1: Flow chart of the SA approach. There are two SA approaches: (Online) RT SA (red flow chart), and (Offline) OP SA (green flow chart). Both approaches use the AXA data; RT SA uses untreated raw data, and OP SA uses treated data. OP SA serves as the reference method. RT SA can be set up either by A. using optimized running parameters obtained by the OP SA, B. using generic parameters that require little to no knowledge of the study area.**

The theoretical foundations for applying PMF are extensively detailed in numerous publications (Manousakas et al., 2021; Paatero, 1999; Paatero and Tappert, 1994). Briefly here, PMF is a mathematical tool used for source apportionment. It decomposes a dataset into a set of factors and their contributions, helping to identify and quantify pollution sources. PMF assumes non-negative values for factors and contributions, making it suitable for real-world environmental data. It allows researchers to trace pollution back to its sources based on the chemical composition and temporal patterns of the collected data. SoFi Pro (Source Finder Pro) by Datalystica was the tool used to implement SA.

SoFi uses the ME-2 (Paatero, 1999) solver, which allows exploring the rotational space around the base solution by introducing limits and/or penalties into the PMF model for deviation from predetermined values for the factor profiles and contributions for one or more factors, a technique called constraining. The implementation of constraints is executed through the a -value approach, in which one or more output factor profiles and/or time series are required to be within predefined limits of a reference profile and/or time series, with the tightness of constraint defined by the scalar a ($0 \leq a \leq 1$). Constraints may apply to the entire profile or time series, or to selected variables and/or time points only. The degree of freedom is regulated by the scalar a (a=0 means 0% allowed deviation from the anchor profile, and a=1 means 100% allowed deviation). Using constraints



in elemental datasets has been shown to provide improved factor separation compared to conventional unconstrained PMF
(Canonaco et al., 2013; Daellenbach et al., 2023; Perrone et al., 2018; Stefenelli et al., 2019).

Additionally, SoFi offers the possibility of implementing a rolling window approach. It has been found in numerous studies
that the rolling window approach, first introduced by Parworth et al (2015), provides better results for organic aerosol SA
compared to the conventional techniques (Bhattu et al., 2024; Canonaco et al., 2013, 2015; Tobler et al., 2021), while to the
best of our knowledge, this technique has not been evaluated yet for high time resolution elemental composition data. The
rolling approach is described in detail under (Canonaco et al., 2021, Chen et al., 2022a) and only a short summary is given
here. This approach involves running PMF on a small subset of the data, referred to as a "window." The process begins with
the window, which is then shifted by a predefined time step, gradually covering the entire dataset. At each step, multiple
individual PMF runs may be performed, with the results either accepted or rejected based on predefined criteria scheme
controlling the quality of the modelled factors. The final source apportionment solution is determined by the set of all accepted
PMF runs.

The specifics for each component SA are presented in the following subsections.

### 2.2.2 Elemental content

For the elemental component of OP SA, all data were used in a single dataset consisting of 515 hourly elemental PM2.5
measurements. The elements selected for the analysis were based on their signal-to-noise ratio (S/N) and their below detection
limit (BDL) values. In the end, 22 elements ranging from Si to Pb were selected for the analysis. To identify the optimum
solution, a number of factors ranging from 3 to 9 were selected. A 5-factor solution was identified as the most environmentally
reasonable and mathematically stable.

The selection of the number of factors in the PMF analysis is identified as the step with the highest uncertainty. In this study,
the optimal solution of the OP SA was determined by combining mathematical diagnostics (e.g., Q/Qexp, scaled residuals,
residual structure, and unexplained variation) with an evaluation of the physical relevance of the factors, based on indicators
such as diurnal variations, correlations with external data, and time series analysis. A range of unconstrained solutions was
initially examined to identify the highest number of factors that could be interpreted with physical meaning. The solutions
were deemed mathematically suitable, when the residuals were found to be normally distributed, unstructured over time, and
consistent across all variables, in line with the findings of (Reff et al., 2007). Rotational ambiguity was addressed using an
approach similar to what it is described by (Canonaco et al., 2021). The a-value method (providing insights into rotational
ambiguity) was combined with the classical bootstrapping method (BS) and newly available perturbation analysis (PR). BS
analysis slightly alters the input by removing some entries and substituting them by repeating other entries. This accounts for
the effect of a small set of observations and random errors in the solution. PR allows for perturbation of the dataset within the
uncertainties of the variables. The uncertainties can be multiplied by a factor; in this case, a factor of 1 was used, meaning that
the values could vary within one reported uncertainty (a factor of 2 would mean double, and so on). The concept is that the
values can range within +/- the uncertainty, with the absolute value being unknown in reality. The model perturbs the dataset



randomly based on the uncertainty or a percentage of the uncertainty (the uncertainty is multiplied with a random value between +/- an user-defined value). The combination of BS with PR and a high number of runs allows for the evaluation of random errors, measurement uncertainty, and the rotational ambiguity of the solution.

In this setup, the number of runs was set to 1000, with the a-value fixed at 1, and BS and PR enabled. With this setting the model performs 1000 runs with slightly altered input at each occasion in terms of number of samples (BS) and absolute concentration of the variables (PR). Since the order of the factors can change with each run, the user must either sort them post-analysis or constrain them to maintain a fixed position. For this reason, all factors were constrained using profiles from the base run (original single run) with an a-value of 1, permitting a 100% variation in the anchor profiles. Using a high a-value

enables the identification of uncertainty without artificially reducing it by not allowing the factors to vary too much from the base case solution. In such cases, the constraint becomes less influential, as each factor can adopt a wide range of shapes. The reported solution is the average of all 1000 runs (no criteria selection used), and the uncertainty was estimated as the variation of the runs, with an average uncertainty of approximately 10% or less across all factors.

### 2.2.3 Organic component

The OP SA PMF analysis of the organic component retrieved four factors, including a hydrocarbon-related OA (HOA), a cooking-related OA (COA), biomass burning (BBOA), and one oxygenated OA (OOA). The initial step of the procedure involved a set of constraint-free runs for a range of factors from 3 to 6. After comparing the 4-factor and 5-factor solution, that varied in the number of SOAs retrieved (one SOA vs two SOAs, respectively), in terms of Q/Qexp, the 4-factor solution was deemed more suitable. Then, the primary organics factors that were identified in the constraint-free runs were constrained one

by one for a series of profile-constrained runs to evaluate the suitability of the constraints. Using the profiles from the unconstrained runs of the specific dataset for the Demokritos station ensures that the constraints are tailored to the site, eliminating potential biases that could arise from relying on profiles from the literature, which may not accurately reflect the conditions of the area. To assess the uncertainty of the solution and further optimize the retrieved profiles, the model was run for 1000 iterations constraining the profiles of the Primary Organic Aerosols (POAs) using the profiles from the previous steps,

with BS enabled using a random a-value up to 0.3. Higher a-values are not suitable for ACSM data, as the identity of the factors can change, in contrast to Xact data. The benefit of this approach is twofold. First, it allows for a comprehensive assessment of the uncertainty in the solution by exploring the variability across multiple runs. Second, it optimizes the retrieved profiles by further fine-tuning them within realistic constraints while ensuring they remain representative of local conditions. From these iterations, only the runs that met specific environmental criteria were selected, further enhancing the accuracy and

reliability of the source apportionment results by ensuring alignment with real-world conditions.

A list of criteria was used to identify environmentally reasonable solutions. This list, based on the methodology described by Chen, Canonaco, Tobler, et al. (2022), included several key elements to ensure the validity of the results. First, a t-test was applied to verify that the correlation of the time series of HOA and BBOA with external tracers of their respective emission





sources (eBClf for HOA and eBCsf for BBOA) was statistically significantly higher than their correlation with other factors.

This ensured that the solutions were physically meaningful and aligned with known emission sources.

Second, the ratio of lunchtime to early morning concentrations of COA was required to be greater than 1, reflecting the expected diurnal variation typically associated with cooking emissions. Lastly, the fractions of m/z 43 (f43) and m/z 44 (f44) for the SOA were required to be positive, as these values are indicative of secondary organic aerosol processes and must adhere to physical and chemical plausibility. These rigorous criteria helped to refine the results by selecting only those solutions that

were consistent with known environmental and chemical behavior, thereby improving the reliability and interpretability of the source apportionment analysis.

In a previous study, (Zografou et al., 2022) identified five organic matter (OM) factors at the Demokritos station in Athens: HOA, COA, BBOA, and two types of oxygenated organic aerosols (OOAs); one more oxidized (MO-OOA) and one less oxidized (LO-OOA). However, in the current study, only a single OOA factor was identified. This finding is supported by the

POA-subtracted f44-f43 plot, which includes the triangle framework proposed by (Ng et al., 2011a). In cases where two OOAs are present, the plot typically exhibits a linear relationship between f44 and f43, which was not observed in this study.

Furthermore, the mass spectrum of the single OOA closely resembles that of the MO-OOA from the 2018 dataset reported by Zografou et al. (2022), suggesting that the OOA identified in this study represents the more oxidized fraction of secondary organic aerosols. The absence of a distinct LO-OOA may indicate differences in atmospheric conditions, such as reduced

variability in oxidation states or changes in the sources and processes affecting the organic aerosol composition between the two datasets, taking into account that this study refers to a single month dataset compared to the year-long dataset used in the previous study. This highlights the importance of site-specific and temporal factors in shaping the chemical composition of atmospheric aerosols.

**2.3 Real time source apportionment analysis (RT SA)**

**2.3.1 Description of the RT model and RT SA approach**

The RT SA was conducted using the SoFi RT software developed by Datalystica Ltd., a tool specifically designed to streamline and automate the source apportionment (SA) process. This software integrates seamlessly with various instruments, as illustrated in Figure 1, to collect raw data in real time. It then applies essential preprocessing steps, such as filtering, blacklisting, corrections, and other necessary adjustments, before performing an automated SA run.

Once the instruments are operational, the software automatically detects when new data are generated and immediately processes them. Within seconds of the data arriving, SoFi RT produces SA results, ensuring a near-real-time analysis capability. The RT model results appear in the form of pie charts showing source contributions and detailed source profiles. These outputs are updated at a time resolution equivalent to that of the instrument used, which is typically hourly or even more frequent. This functionality provides an efficient and dynamic way to monitor source contributions in near-real-time, offering





valuable insights on environmental processes also allowing for rapid decision-making and adaptive air quality management. The RT application eliminates manual intervention, minimizes processing delays, and ensures consistency in the analysis. Once the software is granted access to the relevant instrumental output files, which can be stored on any cloud service, the entire process runs automatically. SoFi RT performs two types of automated source apportionments: a more advanced method (rolling PMF), where multiple PMF runs are conducted to assess errors based on statistical and rotational uncertainty, and a

simpler method using chemical mass balance (CMB) to provide real-time source apportionment results based on the most recent scans. The software can operate with data from individual instruments (or PM components) separately, but it can also process data from the entire AXA setup simultaneously.

To better understand the results of the RT and how to set the model up properly, it is important to know how it operates with the AXA set up. The aethalometer data are decomposed to BC that corresponds to liquid (mainly traffic) and solid (mainly

biomass) fuel combustion. The measured absorption coefficients at wavelengths 470 and 880 nm together with the alpha values based on (Zotter et al., 2017) are used to estimate the contributions to eBC (equivalent BC from eBClf and eBCsf. Moreover, these fractions of BC can be used within the SoFi RT software in order to constrain the solution of the organic factors from the ACSM, by e.g. performing t-tests on the correlation of their time series to the time series of factors of the same emission (HOA and eBClf, BBOA and eBCsf).

The apportionment of ACSM and Xact data is carried out as follows. First, the data from both instruments are automatically combined into two diagonal blocks of one single input matrix. The model then classifies the data from the two instruments into separate classes and retrieves automatically the instrument-specific constraints. The current version allows no interaction between the data from the instruments, hence there will be factors dedicated to either ACSM or Xact data. This information is retrieved from the constraints information (anchor name and length) and is automatically identified and applied. The factor

contributions and profile of the counter-instrument is automatically set to zero, to have independent factor solutions. This method is equivalent to conducting two separate PMF analyses, one for each instrument, however it is performed in one single ME-2 run. The results display all variables (ACSM and Xact) along the x-axis, but the factors for ACSM and Xact are reported separately. The resulting time series of source contributions are separate for each instrument; they consist of a set of time points equal to the rolling window length for the ACSM sources, and an equivalent set for the Xacr. Afterwards, the relative

contributions for the Xact-related data are included for these time points, while the ACSM contributions are set to zero.

The source contributions are presented in a consolidated pie chart that includes the sources of each PM component, though derived from independent PMF runs. This approach has the advantage of providing information about the sources of PM for species representing the entire PM mass. At the same time, it avoids the need to assume equivalent uncertainties for all instruments or to assess whether the model is equally weighting the data from both instruments. The disadvantage is that the

secondary species are not apportioned to sources but rather to oxygenation states, as normally the case for ACSM.

As discussed, the data of the instruments are not combined prior to the analysis. Even though there are other possible approaches, this was deemed the most reliable for RT implementation. A detailed overview of these approaches, along with an evaluation, is provided in the supporting material of (Cheung et al., 2024). The main advantages of the separate run approach





include a straightforward method for estimating uncertainties, low uncertainty for factors with minor contributions, robust
results that align well with RT approaches, reduced impact of apportionment uncertainty on high-mass variables, and easy
implementation by non-experts.

Different analytical techniques typically use distinct methods for estimating uncertainties, and this difference is especially
pronounced in inherently different methods, such as mass spectrometric and spectrometric or other analytical techniques. The
number of variables and the internal correlations within the dataset from a single instrument, typically yields a result in which
instruments are systematically over/under-weighted. Considering that in RT, the treatment of uncertainties is an automated
process managed by a software designed to operate unattended and, in many cases, by non-experts, achieving accurate
uncertainty handling becomes an even more challenging task. By utilizing independent runs, the software can estimate the
uncertainty per instrument using well established and tested approaches by the manufacturer of the instruments that guarantee
robust results.

Since the RT capabilities of the model may be used not only by scientists but also by monitoring networks and policymakers,
the robustness of the results is crucial. By utilizing separate runs and avoiding the need to scale the uncertainties, the results
become very robust. Additionally, including secondary species in the apportionment introduces a limited number of variables
with very high mass into the analysis. Because of this uneven mass distribution, even low uncertainty in the apportionment of
these variables can lead to significant uncertainties in the contributions of the sources. Since these species are typically
apportioned to specific factors associated with secondary sources rather than attributed to primary sources, the loss of
information is not significant and does not justify the added uncertainty in the analysis.

The advantage of the above mentioned set-up, is that for the proper functioning of SoFi RT, the user needs to provide limited
initial information. After that, the model can run autonomously. In this current version of SoFi RT, where the number of factors
is assumed to stay static, the user provides a) general storage settings, b) the total number of factors (in this current version the
factors are fully instrument-independent), c) possible instrument-wise constraints information, c) various other model
parameters. This information can be either passed through an instruction file or directly interacting with the SoFi panels.

### 2.3.2 Organic RT SA

General details on how to set up and monitor the RT runs for the ACSM are provided by (Chen et al., 2022b). Briefly here,
since the RT requires as input the number of sources and an initial set of source profiles, the first step includes identifying this
information. If there is prior knowledge in the specific site, it can be used to set up the initial parameters. If not, seasonal pre-
tests are required to identify the number of factors that are relevant in the region. This procedure is very important, as it will
set up the starting point of the model iterations and it is suggested to follow the analysis protocol described in (Chen, Canonaco,
Tobler, et al., 2022). The POAs are typically constrained using the a-value approach, while the SOAs or OOAs (oxygenated
OA) are left unconstrained. If more than one OOAs are present, the user needs to define additional criteria for repositioning or
sorting the unconstrained factors over the single PMF runs. The model can utilize complementary measurements for validation
of the source apportionment results using the criteria-based selection, as performed manually in the past SA studies.



As discussed earlier, this study aimed to assess the RT model's performance with optimal initial parameters and explore its operation under less optimized, generic conditions. To achieve the first goal, we used OP SA results to set the organic RT SA, constraining POA profiles while leaving OOA profiles unconstrained. The RT was set to perform a rolling window approach
with a-values equal to 0.1, a 5-day window with 1-day shift, 50 repeats per window and bootstrapping enabled. This setting allowed the model to adjust the profiles to temporal changes. The window size was chosen because the dataset is small, allowing for enough subsets and iterations. For longer datasets (e.g., 1 year), a more traditional method like the one in (Canonaco et al., 2021) is recommended. The evaluation under optimized conditions aims to assess the stability of the model, the processing of the raw files, and the overall performance.

For checking the performance of the model under non-optimized set up, we used as constraints the most commonly used profiles worldwide based on the studies of (Crippa et al., 2013; Ng et al., 2011b). These profiles are used in numerous studies to constraint the POAs (Chen et al., 2022a). Generally, these profiles work well for HOA and COA because their fingerprint is relatively consistent across different locations. HOA is typically associated with traffic emissions, which have similar chemical characteristics globally due to the widespread use of similar fuels and combustion technologies. Similarly, COA
profiles are dominated by cooking emissions, which also exhibit comparable chemical signatures worldwide, driven by commonalities in cooking practices and the types of oils and fats used.

In contrast, BBOA profiles are less consistent globally, as they are highly dependent on the type of biomass burned, the combustion conditions, and regional variations in vegetation. Different types of wood, agricultural residues, or other biomass materials produce unique chemical markers during combustion. Additionally, variations in burning methods (e.g., open fires,
stoves) and atmospheric conditions can further alter the BBOA profile. As a result, using a generic global profile as a constraint for BBOA may not capture the local and regional variability, leading to less accurate modeling outcomes for this source.

The performance of both approaches was evaluated by comparing the results of the RT SA to that of the OP SA and the results are presented in a following section.

### 2.3.3 Elemental RT SA

Even though the RT function for the OA has been previously evaluated, its application on Xact data has not yet been assessed. Since in the RT model the source profiles and the number of sources are fixed parameters, the performance of RT can be significantly affected by two main factors: variations in the chemical composition of source profiles and changes in the number of sources. Unlike the OA data, where predicting the number and type of sources is relatively easier, often allowing for the same profiles to be used as starting points independent from time and location, elemental composition data are much more
unpredictable. While certain factors with similar chemical compositions are common on a global scale (e.g., dust, salt, secondary species, and possibly traffic), many others show significant variation depending on dominant local processes such as industry or regional pollution patterns. Even in cases where sources share similar tracer profiles, like dust for example, the relative concentrations of variables within those factors can change based on the location and the type of dust (e.g., local natural, local anthropogenic, or transported natural dust). Additionally, transient sources frequently appear, which can have a



substantial impact on elemental concentrations, in case they become part of the main modelled factor solution. These sources are often short-lived and cannot be predicted at the beginning of the analytical cycle. Even though there are studies that suggest optimized SA approaches for offline SA analysis of Xact (Manousakas et al., 2022), they are quite complicated to operate under RT conditions.

For the optimized RT SA the source profiles retrieved from the OP SA were utilized in this case as well. The number and types

of sources identified by the OP SA were used as initial parameters to run the RT model. The model's ability to adapt to potential changes in factor profiles relied on the selected a-value, which allowed for adjustments to the profiles when combined with the rolling window approach. An a-value of 0.5 was chosen, enabling the model to modify the profiles by up to 50%. Higher a-values (even 1) are possible for the Xact, as the zero values for some variables, which almost always exist, help maintain the profile's identity more stable compared to what is usually the case for the ACSM. This approach provides the model with

flexibility to adjust the profiles while maintaining alignment with the initial parameters. The success of this approach heavily depends on the quality and representativeness of the initial parameters. To ensure optimal performance, locally derived OP SA data must be available.

While the previous method is effective in delivering robust results and offering the model some flexibility to adapt to changes, it requires prior knowledge of the specific area being studied. Increasing the a-value further enhances the model's adaptability,

but there are limitations on how high it can be set without introducing modeling issues. For instance, setting an a-value of 1 provides significant flexibility by allowing all variables to potentially reach zero. However, variables that are already at zero remain fixed and cannot be adjusted, which can limit the model's ability to capture certain dynamics accurately.

An additional limitation is that although all variables can theoretically reach zero, the maximum relative increase permitted is only 100% (i.e., doubling their current value), which may be insufficient for variables with low initial concentrations.

Furthermore, allowing variables to reach zero can result in the model unintentionally altering the identities of factors. This occurs when the model compensates by redistributing species into other factors, prioritizing those with higher initial concentrations that align with the zeroed variables. Consequently, this process can lead to factors effectively exchanging their "identities," undermining the consistency and reliability of the source apportionment results. To account for that, these exchanged/mixed PMF runs can be effectively filtered out using proper (ideally based on statistical tests) thresholds within the

criteria scheme.

The high effect that the zero values in the PMF input matrix have on the solution is also described in previous publications (Paatero et al., 2002). The presence of zeros in profiles can play a crucial role in maintaining factor identity, especially when a significant number of variables are set to zero for each factor and there is minimal overlap in the zeroed variables across different factors. These zeros effectively anchor the factors, preventing ambiguity and ensuring their distinctiveness (Table.

S1).

Building on this observation, we developed a novel approach for creating constrained profiles. This approach involved setting certain variables to zero (those deemed irrelevant for factor identification) while allowing random initial values to the remaining variables. This method allows the model to determine the relative concentrations of variables within each factor



without imposing any predefined assumptions about their relative composition. By doing so, the model gains the flexibility to

adapt and allocate variables freely based on the data, ensuring that the profiles remain data-driven and unbiased.

This "zero/nonzero" strategy was tested to evaluate its ability to provide sufficient information for the model to accurately identify relevant factors. Simultaneously, it removed constraints on the relative concentrations of species, offering a more adaptable framework for the model. The approach aimed to strike a balance between maintaining factor identity through zero constraints and enabling a versatile, unconstrained exploration of the data to optimize the accuracy and reliability of source

apportionment results.

The performance of both approaches was evaluated by comparing the results of the RT SA to that of the OP SA and the results are presented in a following section.

## 3 Results

### 3.1 OP SA results

To evaluate the performance of the RT model, its results were compared with those obtained using the classical offline approach to source apportionment (SA). The outcomes of each approach are presented in separate sections for each instrument. For clarity, the results of the OP SA are detailed extensively in the following sections, while the RT SA results are presented primarily in comparison to the OP SA outcomes.

### 3.2 Elemental component SA results

Following the SA approach described in the Methods section, six factors were identified utilizing the elemental composition data. The factor profiles are presented in Figure 2.





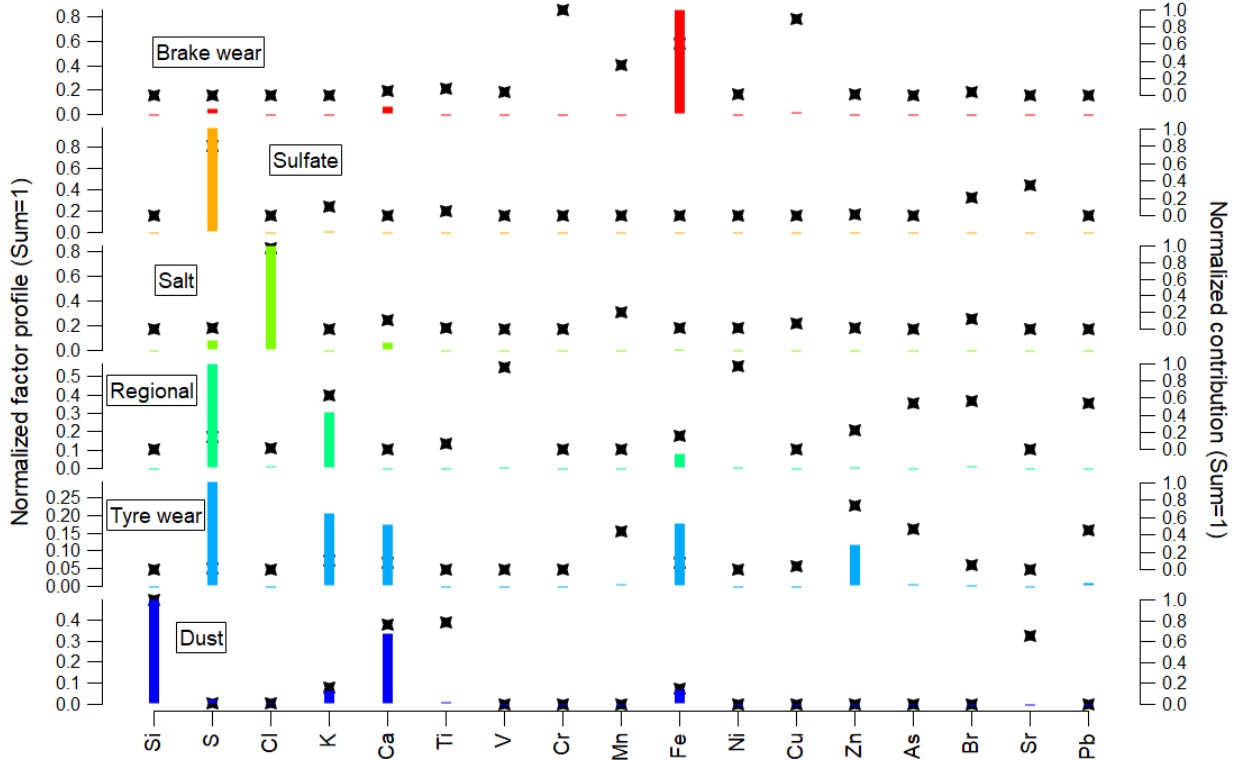

**Figure 2: Factor profiles for the elemental SA. The bars represent the normalized (Sum = 1) factor profile, while the stars represent the normalized (Sum=1) contribution of the factor for each species.**

### 3.2.1 Dust


Dust is traced by Si, K, Ca, Ti, Mn, and Fe. Greece, as the rest of the south European countries, is often affected by dust transport events. Mineral dust, regarding its origin, has somewhat different composition, with the main difference being the relative content of Al, Si, and Ca. Elements such as Fe and Ca, originate strongly from transported natural dust, but also have significant local emissions either from anthropogenic dust emissions such as construction and the abrasion of building materials

and road surfaces (Ca), or traffic emissions (Fe) (Amato et al., 2013; Nava et al., 2012; Shaltout et al., 2018). For the aforementioned reason, dust sources in certain regions may appear as two distinct sources, one which is Ca dominated and represents local/urban dust or construction, and one that is Al and Si dominated and represents natural and/or transported dust (Manousakas et al., 2021). Ratios of Al and Si to Ca can be used to distinguish between anthropogenic and natural transported dust (Shen et al., 2016; Skorbiłowicz and Skorbiłowicz, 2019). The Si/Ca ratio in the factor profile is approximately 1.5,

indicating that the factor is affected by both natural and anthropogenic emissions, which is common in urban environments. The time series of source contributions for this source reveals some events that are attributed to long-range transportation events (Fig. S3).



### 3.2.2 Traffic related factors

In the analysis two factors that refer to vehicular traffic emissions have been identified; one factor refers to emissions from
brake wear, and one factor refers to emissions from tyre wear. Since we are focusing on the elemental component of PM, both
correspond to non-exhaust emissions, as exhaust emissions contain mostly carbonaceous species (Harrison et al., 2021). Brake
wear factor is traced by Cr, Mn, Fe, and Cu. Cu is the most abundant element in brake lining having a concentration that
reaches >10%, while Cr, Fe, and Mn can also be emitted from braking as they are components of brake linings and/or the brake
disc/drums (Thorpe and Harrison, 2008). The tyre wear factor is traced by Zn, Mn, As, and Pb, while Fe, Ca, and S contribute
significantly to the mass of the factor. Although tyre wear is predominantly associated with the release of organic compounds,
approximately 13% of tyre composition consists of inorganic materials, such as those found in curing agents, accelerators, and
various additives (Thorpe and Harrison, 2008). Additionally, several trace metals, including Cd, Cu, Pb, and Zn, are used in
tyre manufacturing. Among these, zinc has the most substantial presence in tyre tread, comprising about 1% of its total weight
(Kleeman et al., 2000). While the chemical compositions of tyre wear and brake wear do have some differences, their temporal
emission profiles provide a strong basis for differentiating. These time-based variations help distinguish the two sources in
real-time PM monitoring, as each has distinct peak emission times linked to driving behavior, road use, and traffic conditions.
Tyre wear tends to be more continuous throughout the day, as tyres are in contact with the road surface during any kind of
driving; while it is suggested that particles from tyre wear are elevated during higher driving speeds (Gustafsson et al., 2008;
Kim and Lee, 2018; Yan et al., 2021). Brake wear emissions are more sporadic and directly related to braking intensity, which
typically increases during rush hours or in areas with stop-and-go traffic. Consequently, brake wear peaks during periods of
heavy congestion, especially in urban areas where frequent braking occurs, leading to higher PM contributions during morning
and evening rush hours. In traditional source apportionment (SA) analysis, which relies on 24-hour filters, there is often
insufficient variability in the data to separate brake wear and tyre wear, resulting in both being grouped as a single source.
However, with hourly resolution data, these sources may be differentiated. The diurnal profiles of brake and tyre wear are
notably distinct: brake wear exhibits clear rush hour peaks, with noticeable spikes in the morning (around 07:00) and evening
(around 19:00), while tyre wear shows a more irregular profile, with a less pronounced peak around noon (12:00), when traffic
density is lower, and vehicle speeds are higher (see Figure S1 in SI). Traffic data from the nearby highway corroborates these
findings, indicating lower traffic density at 12:00 and higher density between 07:00-09:00 and 17:00-19:00, which aligns well
with the observed profiles for both factors (see Figure S2 in SI).

### 3.2.3 Salt

Salt factor is traced by Cl and refers to sea salt, as road slating rarely takes place in Greece and especially during spring. The
interesting observation about this source is that it presents pronounced peaks during the same time as the dust transport events
(Fig. S3). Sea salt can be transported with African dust, especially during large-scale dust storms that originate from the Sahara
Desert. The process involves strong winds lifting both dust particles from the desert and sea salt from the ocean surface into



the atmosphere. These particles can travel long distances together, becoming part of a mixed aerosol layer in the atmosphere (Van Der Does et al., 2016; Goudie and Middleton, 2001). The mass concentrations of PM1 and PM2.5, as measured by an Optical Particle Counter (OPC, GRIMM 1.109), are shown in the Supplement (Fig. S4). Periods during which chloride appeared in the sea salt factor identified by the Xact monitor but was not detected by the ACSM, since the chloride contained in sea salt is refractive and not captured by the ACSM. Moreover, these peaks corresponded to instances where the PM2.5

mass was significantly higher than the PM1 mass.

### 3.2.4 Sulfate

Sulfate is traced by S, and represents the secondary sulfates in the region. Sulfate has been identified in the past as an important source in the region (Almeida et al., 2020; Amato et al., 2015, 2016). The area's climate conditions, characterized by low precipitation and high solar activity promote the buildup of pollutants and the generation of secondary particles. For instance,

model simulations suggest that SO2 is carried throughout the Mediterranean basin, where sulfate is formed as a result of significant photochemical activity (Pikridas et al., 2013). Sulfates have been found to have similar concentrations in several areas in the Mediterranean region (Argyropoulos et al., 2012), highlighting the regional character of these secondary aerosol species.

### 3.2.5 Regional

This factor represents regional pollution that is transported to the sampling site from most likely outside of the city. The factor includes tracers from heavy oil combustion (V and Ni), biomass burning (K), as well as industrial processes (S, As, Br, and Pb) (Jang et al., 2007; Samara et al., 2003; Sánchez-Rodas et al., 2007). In addition to local emissions, Athens is affected by industrial activities in nearby areas and shipping emissions from the Port of Piraeus. Factories involved in manufacturing and petrochemical production contribute significant emissions that can be transported to Athens by prevailing winds. Emissions

from heavy oil combustion and industrial activities are often grouped together in a source apportionment factor due to synchronous transportation mechanisms that result in their simultaneous presence in the atmosphere, making it difficult for models to effectively distinguish between sources. The unique geographic and meteorological conditions of Athens, including its location in a basin surrounded by mountains, create an environment where pollutants can become trapped. Prevailing winds can carry emissions from both heavy oil combustion and nearby industrial sources into the city at the same time, leading to

overlapping pollutant plumes that complicate the identification of specific sources. Furthermore, both heavy oil combustion and industrial activities release a variety of pollutants with similar chemical compositions. This overlap makes it challenging to attribute specific air quality issues to a single source. Instead, a composite source apportionment factor is employed to assess the total impact of these emissions on air quality.



### 3.3 Organic matter (OM) offline SA

The mass spectrum of the identified factors is presented in Figure 3, which covers an m/z range up to 100. Higher masses, often associated with polycyclic aromatic hydrocarbons (PAHs), showed minimal contribution to the spectra in this case. The two factors related to hydrocarbons, HOA and COA, are characterized by peaks at m/z 41 and 55, which are indicative of alkanes, and at m/z 43 and 57, also representative of alkane fragments (Zhang et al., 2005). These fragments are crucial markers for identifying emissions from vehicle exhaust (HOA) and cooking sources (COA).

A distinctive feature differentiating HOA from COA is observed in the ratio of the m/z 55 to m/z 57 peaks. In the case of COA, this ratio is greater than one, which is indicative of a predominance of lighter alkanes typical of cooking emissions (Mohr et al., 2012). In contrast, HOA displays a ratio lower than one, reflecting the heavier alkanes often associated with vehicle emissions.

The BBOA factor stands out due to a prominent peak at m/z 60, which is attributed to levoglucosan (and similar biomass

burning related compounds), widely recognized as a biomass burning tracer. This peak is a good marker for identifying organic compounds from wood or crop burning.

The OOA factor, meanwhile, is predominantly represented by a strong peak at m/z 44, which corresponds to the $CO_2$, a marker for oxidized organic compounds, especially acids (Duplissy et al., 2011). This ion is a product of atmospheric oxidation processes, such as those occurring in SOA formation. The prevalence of m/z 44 is typically associated with the oxidation of

organic precursors in the atmosphere, reflecting the processing of primary emissions into secondary aerosols (Kanakidou et al., 2005).



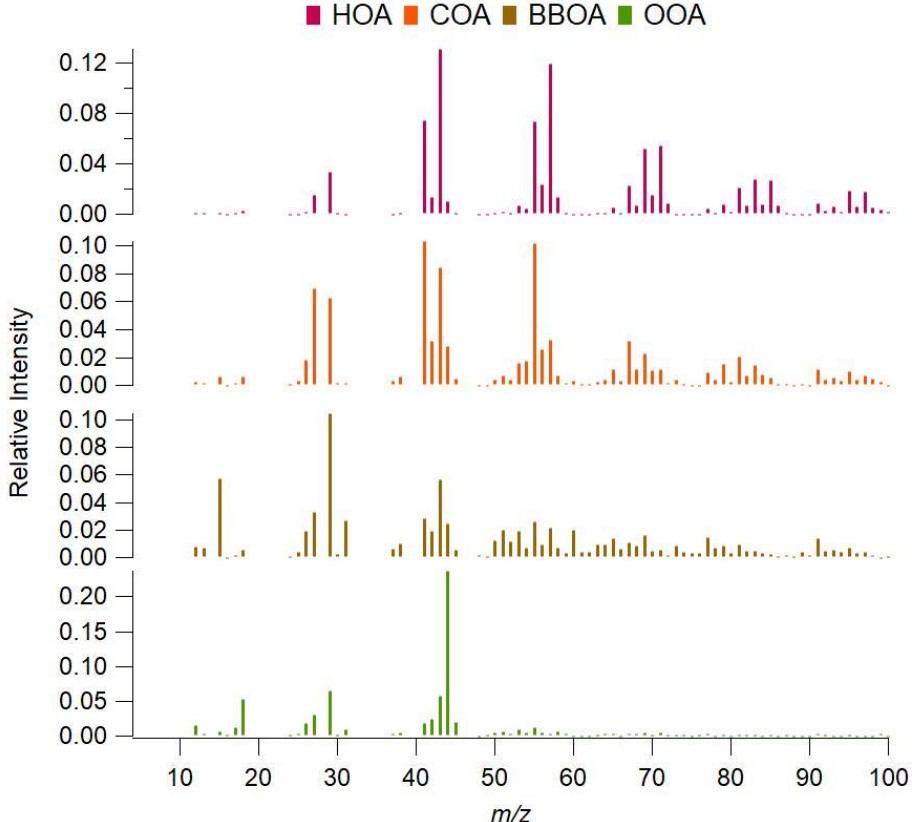

**Figure 3: Mass spectrum of the best-case of organics factors**

The time series of the ACSM factors retrieved from RT PMF appear in the SI (Fig. S5). An interesting event occurred on the

7th of March, when an annual Greek festival centered around meat grilling took place. During this event, there was a significant spike in COA contributions, which was accompanied by a notable increase in BBOA concentrations. Figure S6 presents the factors diurnal trends. HOA concentrations were observed to peak twice per day, reflecting vehicle emissions, while COA also presented a bimodal diurnal trend concentrations coinciding with lunch and dinner times. BBOA showed a pronounced evening peak, probably driven by the grilling event on 7th of March.

When comparing the base case solution factors to external data, correlations were found between the HOA factor and eBClf, with a Pearson correlation coefficient of 0.61. This suggests a moderate relationship between HOA and liquid fuel-related particles. On the other hand, the BBOA factor showed a strong correlation with eBCsf, with a Pearson coefficient of 0.86. This strong correlation underscores the close association between biomass burning sources and the BBOA factor.





### 3.3.1 OP SA source contributions

The results presented here are derived from the OP SA but are formatted to be equivalent to the output provided by SoFi RT. During RT operation, the pie chart of source contributions is updated within seconds after new data from the instruments are processed by the model, which typically occurs every hour or less.

When using the combination of AXA instruments in SoFi RT, the software generates a real-time consolidated pie chart that includes sources from all individual analyses performed. However, in the version used in this study (SoFi 9.5.4), all source

contributions are reported separately in RT mode. While offline mode offers additional options, such as combining equivalent sources, including ions in the pie chart, and adjusting dust contributions for the mass of corresponding oxides, these features were not available in RT mode at the time the study was conducted. However, they are included in the latest available version. Consequently, the pie chart generated in RT mode represents only the apportioned mass of the analyzed species, rather than the total PM mass. Future software updates will address these limitations, providing users with greater flexibility to customize

the graphical representation of the data according to their needs.

Figures 4 and 5 present the source contributions from the OP SA in a format equivalent to what SoFi RT reports during real-time operation, ensuring consistency with the RT reporting style.

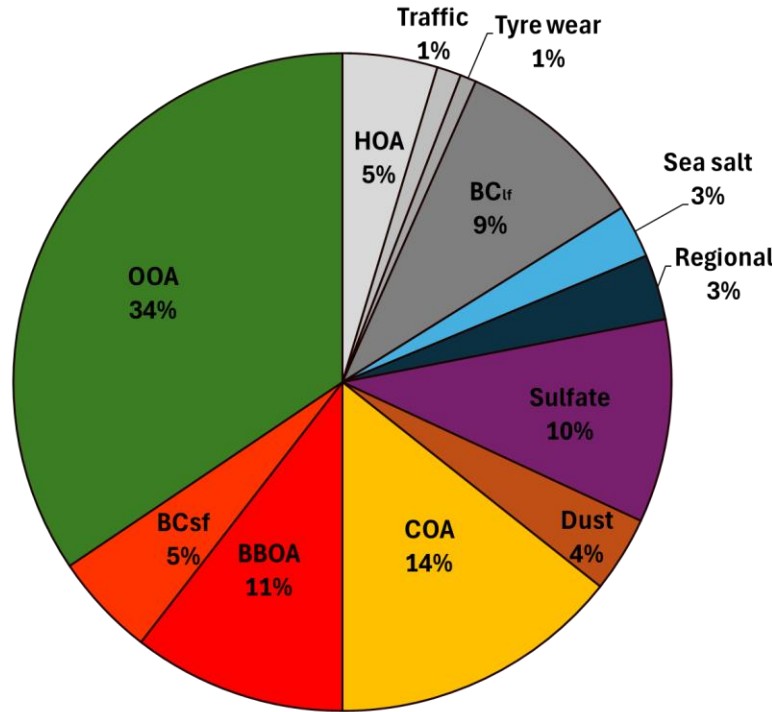

**Figure 4: Consolidated pie chart of the average source contributions. The total mass does not include ions; dust is not adjusted for**
**the oxides; sea salt is not adjusted for the missing Na**



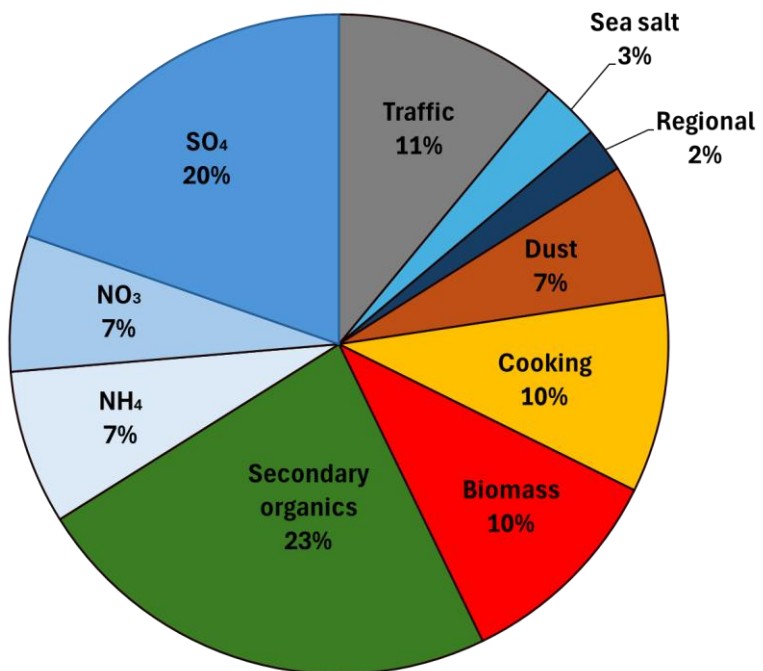

**Figure 5: Consolidated pie chart of source contributions. Traffic is the sum of HOA, Break wear, Tyre wear, and BClf; Biomass is the sum of BBOA and BCsf; Dust is adjusted for the oxides; Sea salt is adjusted for the missing mass of Na; Sulfate factor from the**
**Xact is not included to avoid double mass counting; Chlorine from the ACSM is not included to avoid double mass counting**

In Fig. 4, contributions from the same source category are represented using different shades of the same color to enhance visualization and differentiation. The pie chart in Fig. 5 includes ions, adjusts dust to account for oxides, and incorporates the missing mass of sodium (Na) in sea salt to provide a more comprehensive representation of source contributions.

The largest contributors are secondary species, which together account for 57% of the total mass. Among these, secondary
organics are predominant, followed by sulfate, nitrate, and ammonium. Notably, sulfate as measured by the ACSM shows excellent agreement with the corresponding factor in the elemental SA analysis (1.24 μg/m³ from the ACSM vs. 1.28 μg/m³ from the elemental SA). This consistency underscores the reliability of the analysis.

Since the site is classified as an urban background site, high concentrations of secondary species are expected, consistent with previous findings (Eleftheriadis et al., 2021).

Among primary sources, Traffic emerges as the highest contributor. This category encompasses HOA, BClf, and Break and Tyre wear identified in the elemental SA. Biomass Burning and Cooking sources each contribute 10% of the total mass. The contribution of Cooking, however, is exceptionally high for this dataset and not representative of typical regional conditions. This elevated contribution is attributed to a national celebration day included in the sampling period, during which widespread barbequing occurs. Similar high contributions from cooking during such events have been reported in other local studies
(Manousakas et al., 2020).





Natural sources, including Sea Salt and Dust, together account for 10% of the total mass. These factors are explored in detail in the elemental SA section and are linked primarily to some transportation events that occurred during the sampling period. Finally, the Regional factor contributes 2% of the total mass. This factor is attributed to pollution transported from surrounding areas, as discussed in the corresponding sections.


### 3.4 Comparison of the RT results to best case SA analysis

In Figures 6 and 7, the ratios between the OP SA application, as described in the previous section, and the RT implementation of the a-value approach are presented. Based on this comparison, the RT model application results appear robust, with most ratios being close to 1 for both instruments.





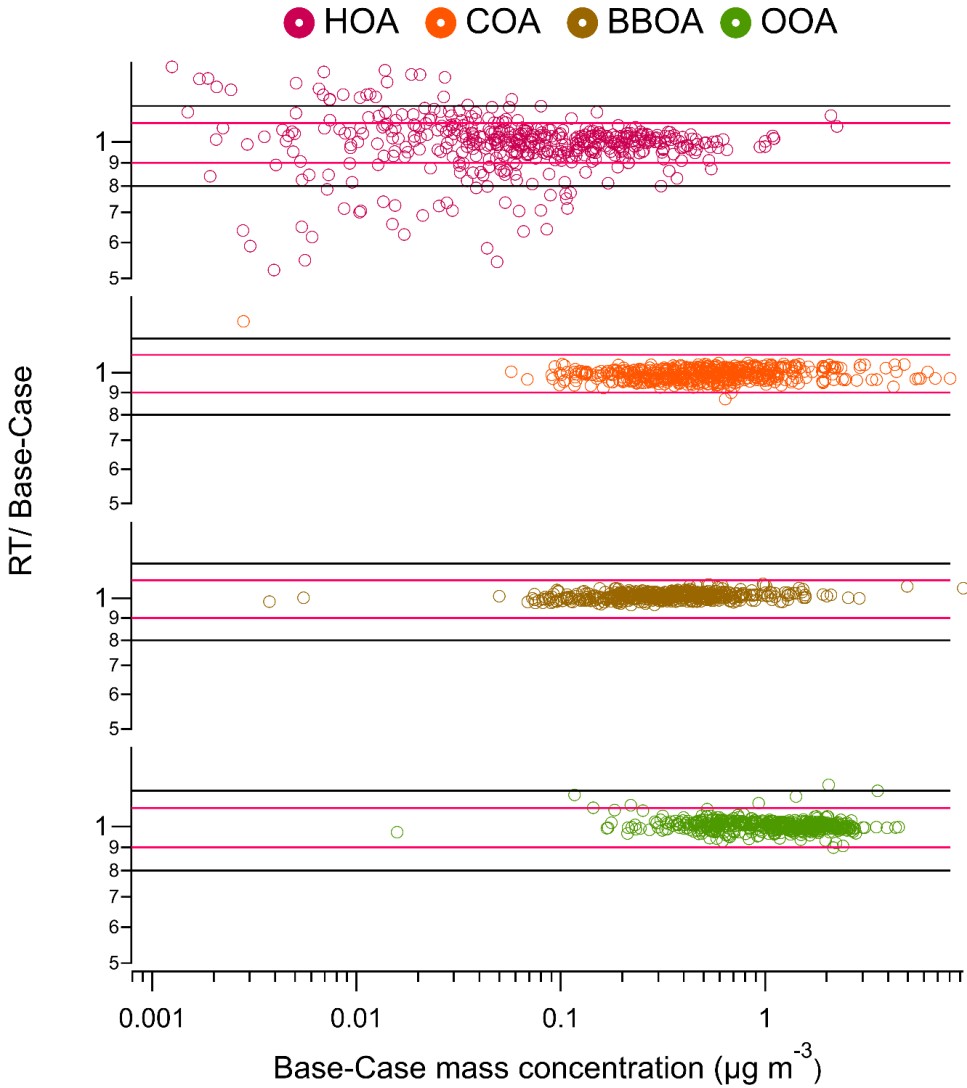

**Figure 6: Ratio of the source contributions of the RT SA analysis to the to the base-case offline SA application vs the source contribution in μg m-3 for the organics based on offline SA, with lines indicating the ranges ±10% from 1 (red) and ±20% from 1 (black).**







**Fig 7. Ratio of the source contributions of the RT SA analysis to the to the base-case offline SA application vs the source contribution in µg m-3 for the elements-based offline SA with lines indicating the ranges ±10% from 1 (red) and ±20% from 1 (black).**

To better visualize the quality of the RT solution compared to the OP SA solution for both instruments, the lines indicating the ±10% (red) and ±20% (black) range from 1 in the ratio of RT/Base-case were added in Figures 6 and 7. In quantitative terms, 98 % - 100 % of the data were inside the ±10% range in the case of COA, BBOA and OOA for the ACSM solution. For HOA, lower percentage was observed within this range, and 84% was inside the ±20% range. Concerning the elemental RT solution, the ratio of RT to Base-case for the traffic-related factors, brake wear and tyre wear, was 63% and 77%, respectively inside



the range ±20%. Lower percentage inside the ±20% range was observed for the Sea salt factor (57 %), while the highest percentage was shown for the Sulfate factor (90 %). Finally, 87 % and 81 % of the regional and dust factors ratios, respectively were within this range.

The comparison shows that the highest variability exists for low concentrations and for factors that either are characterized by high intensity events (sea salt) or have a high common number of tracers with other factors (HOA, Regional, Traffic). Regarding Sea salt, the factor has two unique features; it is traced by only one element, and it is expressed by high intensity events. For these reasons, the factor is more sensitive to the BS runs the stability of the results is affected in the windows that include the high intensity events. This effect might be mitigated by increasing the repeats per window and by disabling the BS

analysis. In general, the results indicate that if site-specific profiles are used, the RT analysis can offer results that are practically identical to the best-case offline SA analysis.

As discussed in the previous sections, the implication with using site-specific constraints is that there needs to be prior knowledge in the area, or an initial test period needs to take place. Even though the software offers ways for the model to adapt to changes (rolling window, adjustable a-value), there is still the need for a relatively good and robust starting point in terms

of used profiles.

For the implementation of the elements-based RT-SA, the zero/non-zero approach was also used. This approach includes setting the variables that are not relevant to the factor to zero, while leaving the others free to assume any value. The profiles that have been used to test this approach are presented in the SI (Table S1). The results of this approach were satisfactory presenting very good reproduction of the best case solution regarding the $R^2$ Pearson correlation of the time series of the

contributions of the solutions (traffic= 0.88, sulfate=0.94, sea salt=0.99, dust=0.99), moderate for the regional factor (0.64), and very low for the tyre wear (0.04). In the SI the scatter plots between the diurnal trends for the two approaches are also presented (SI Fig. S7). The $R^2$ correlations in this case were lower, ranging from 0.18 to 0.7. From these results it can be assumed that the factors that have low mass do not follow a consistent pattern; some factors can be reproduced (e.g. regional), some factors can be reproduced only in some cases, while others cannot be well reproduced. Since in this zero/non-zero

approach there are no certain ratios that are fixed between elements, when the factors have high number of overlapping elements, then the model can swap them assigning the mass of one factor to the other (mostly favoring the factor with the higher mass). This is supported by the two slopes that are visible in the scatter plot between the offline base case solution and RT (SI Fig. S8). Overall, although the performance of this approach is not exceptional, it can serve as a supplementary tool in situations where prior information about the sources is unavailable. It can be utilized until sufficient data are collected to enable

an OP SA evaluation.

The performance of the RT SA for the organic fraction was also evaluated by comparing the use of reference profiles to the use of optimized profiles derived from local data. The time series generated by the RT SA using optimized profiles were compared to those generated using reference profiles, revealing varying levels of correlation depending on the source.

For COA, the time series showed a strong correlation ($R^2$=0.81) (Fig. S9), indicating that the reference profile adequately

represents the cooking organic aerosol emissions at the study site. This result suggests that COA has relatively consistent



characteristics across locations, making reference profiles effective for this source. In contrast, the correlation for HOA was moderate to low (R2=0.48) (Fig. S9). This indicates that while the reference profiles capture some general trends, they fail to fully represent the local variability in traffic-related emissions. Factors such as differences in fuel composition, vehicle types, and driving conditions may contribute to this discrepancy. The BBOA time series showed very poor correlation (R2=0.20)

(Fig. S9) between the RT SA using the optimized BBOA profile and the RT SA using the reference profile. This result highlights the spatial dependence of BBOA characteristics, which are influenced by factors such as the type of biomass burned, combustion practices, and local atmospheric conditions. Reference profiles for BBOA appear to lack the specificity needed to accurately reflect the unique features of emissions at the study site as has been previously observed (Chen et al., 2022).

## 4 Conclusions

This study successfully demonstrated the integration of the ACSM-Xact-Aethalometer (AXA) setup with the SoFi RT software for real-time source apportionment (RT-SA) of particulate matter in Athens, Greece. The findings underscore the potential of real-time methodologies in advancing air quality management, offering near-instantaneous insights into pollution sources and enabling dynamic responses to pollution events.

The AXA setup proved effective in providing a comprehensive representation of PM sources. By integrating chemical,
elemental, and black carbon data, the system has the ability to capture most of the PM mass, allowing for detailed source characterization. The analysis identified traffic emissions as the dominant primary source of PM, with substantial contributions from secondary species (57% of the total PM mass) such as secondary organics, sulfate, nitrate, and ammonium. Other primary sources such as biomass burning and cooking each contributed approximately 10% to the total mass, with natural sources like dust and sea salt accounting for the remainder. The consistency of these results across RT and offline analyses demonstrated
the robustness and reliability of the RT methodology.

The results highlighted the diurnal patterns of specific sources, with traffic-related emissions peaking during morning and evening rush hours, while cooking emissions spiked during weekends and special events. Additionally, the setup's ability to differentiate between non-exhaust traffic emissions, such as brake and tyre wear, provided valuable insights into source profiles.

A significant methodological insight from this work is the approach to combining data from the AXA setup. While further testing on combining AXA data prior to source apportionment analysis is warranted, this remains an ambitious goal given the current early stages of applying RT techniques. Combining data prior to analysis introduces challenges, such as differences in data uncertainties and variable correlations across instruments, which can lead to biased weightings or overrepresentation of certain components. By maintaining separate analyses and combining results during post-analysis, the unique strengths of each
instrument are preserved, uncertainty propagation is reduced, and a more balanced attribution of sources is achieved.

The performance of the real-time source apportionment (RT SA) was evaluated under two distinct scenarios, highlighting the flexibility and innovation of the methodology. In the first scenario, RT SA results were compared to those of the optimized

offline source apportionment (OP SA). This comparison demonstrated that when site-specific constraints are used, the RT model can deliver results closely aligned with the OP SA, showcasing its robustness and reliability under optimized conditions.

In the second scenario, the RT SA performance was evaluated using the novel zero/nonzero approach for the elemental data and reference profiles for the organic data. The zero/nonzero strategy represents an innovative method of profile constraint, selectively setting variables irrelevant to factor identification to zero while allowing others to vary freely, enabling greater adaptability to local and temporal conditions. For the organic data, the use of reference profiles highlighted the challenges associated with spatially dependent factors, such as BBOA, compared to globally consistent factors like COA. These

evaluations underscore the versatility of the RT approach, demonstrating its capacity to perform well under optimized conditions while providing a viable alternative when prior site-specific information is unavailable.

The application in Athens illustrates the practical utility of RT-SA techniques in complex urban environments, where diverse pollution sources and fluctuating conditions necessitate advanced monitoring capabilities. The outcomes of this study provide a foundation for improving air quality management strategies and developing targeted interventions to reduce pollution

exposure effectively. Future studies are essential to further evaluate the stability and performance of the RT SA over extended monitoring periods and under varying environmental conditions. Long-term studies will provide deeper insights into the model's ability to adapt to seasonal variations, transient sources, and evolving source profiles. Additionally, exploring alternative ways to utilize the AXA data, could significantly enhance its application in air quality management.

**Acknowledgements**

This project has received funding from the European Union's Horizon Europe programmes under grant agreement No 101138449 — MI-TRAP and under grant agreement 101036245 — RI-URBANS.

**Data availability**

Data are available upon request to the author (m.manousakas@ipta.demokritos.gr)


**Author contributions**

MIM and OZ performed the formal analysis and wrote the original draft. KE, and ASHP helped with funding acquisition, project administration and resources. All authors contributed to data acquisition. MIM, and OZ performed the investigation and data curation. MIM, OZ, FC, and ASHP provided the methodology and conceptualization. FC and AT developed and

provided the SoFi RT Software. KE, and ASHP provided supervision and validation. All authors contributed to reviewing and editing the manuscript.

**Competing interests**



Francesco Canonaco and Anna Tobler are employed by Datalystica Ltd., the official distributor of SoFi Pro and SoFi RT
licenses.

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
