# Peer review of "Implementation of Real-Time Source Apportionment Approaches Using the ACSM-Xact-Aethalometer (AXA) Set-Up with SoFi RT: The Athens Case Study"

_EGUsphere, 2025_

## Referee Comment (RC1)

The paper by Manousakas et al. uses aerosol composition data obtained online at high-time resolution with an ACSM, AE33, and Xact to determine sources of aerosols in Athens during a month- long study. The source apportionment (SA) approach is carried out with SoFi offline and in real-time to demonstrate performance of the real-time (RT) SoFi software when combining data from these 3 instruments. Based on the statement in the abstract, this is the first study investigating combination of all these data in SoFi RT. I think the results and approach are interesting and show promise for using SoFi RT with data from ACSM and Xact. I have some concerns and would like the authors to clarify a few details before accepting the paper.

Major comments:

1. Although from the abstract I got the impression that SA was applied on all the data together, it seems given the different levels of uncertainties and variable magnitude of species concentrations from each measurement =, the data are actually treated separately, by running two independent PMFs in one ME-2 experiment. So is that really combining data to do source apportionment? I understand that because of the reasons explained in the paper, it may not make sense to include secondary aerosol species in the PMF run of primary metal components, but then I think the language in the abstract/body of the paper/conclusion needs to be changed. I think the paper also needs to better explain what is the advantage of running both PMFs in one ME-2 run rather than separately (other than more manual work at the end to combine the results if ran separately).

2. This is a more philosophical comment, but I'd like to hear the authors thoughts on this. As shown in Fig. 5, secondary OA, sulfate, primary HOA, BC, BBOA are a large fraction of PM1. Of course this is a picture for Athens only, but I think in most other urban areas this is still the case. If I understood the conclusions correctly, gaining a valid representation of factors that can have variable profiles depending on time of year or location by RT SoFi requires some knowledge about their actual profile before running RT SoFi. So what is the advantage of RT SoFi if one needs to also run SoFi offline to gain that initial knowledge and how often should the offline SoFi be done if we think profiles of some factors changes with time (say changes in fleet, fuels, etc)?

Technical comments:

1. Sampling section: Although this is an analytical method paper, types of inlets used with each instrument should be included (PM1 cyclone, etc).

2. What was the time resolution of AE33? Also, are the data from AE33 (BClf, BCsf) just used in the last step when combining concentrations of different PM factors/species? L149 indicates PMF was used on AE33 but I don't see eBC showing up in any of the Xact or ACSM profiles.

3. Why wasn't Ba used as one of the main Xact species? Ba is a good marker for brake wear.

4. L194: what threshold for S/N of elements was considered?

5. L228: typically more information from PMF results are included in SA papers to justify the choice of specific solutions. Can they authors add for example Q/Qexp plots as a function of factor number?

6. L236: can you please explain why a higher 'a-value' would affect the results for ACSM but not Xact?

7. L252: although it's mentioned later in the paragraph that the reference study was for a year-long, I think that information needs to be mentioned in the beginning of this sentence.

8. L330-331: what other model parameters are needed to be defined?
9. L406- starting in this paragraph is where the zero/non-zero approach for constraining the profiles is discussed. It's unclear still to me what basis is used to use the 'random values'. Why are the values for the same element for the same factor not the same in the base run and RT run? I'm still very confused how this approach sets the relative concentrations but not the relative composition.
10. What interpreting the Regional sources, I think a better use of WD and location of sources can be helpful. Can a map and wind roses be drawn. Is there a specific direction that has the heavy oil/industrial sources vs. port emissions?
11. L526: during the Greek Festival, COA, HOA and BBOA all increased. Does it mean that HOA is also not separated well from BBOA/COA?
12. Fig 4: what is the traffic factor? Is the Brake Wear? If so, rename it for clarity
13. Fig. 5: I find this pie chart a bit confusing/misleading. NO3- and SOA and maybe even NH4+ all likely have contributions from traffic-related emissions (NOx, VOC, and NH3 emissions). Perhaps a better categorization would be "primary traffic" instead of "traffic". Biomass should be "Biomass burning".
These clarifications still don't fix the issue of mixing sources with composition. I think the best is to not mix these two concepts and actually have individual pie segments for each composition category (i.e., a segment for HOA, brake wear, etc.).
14. Fig 5: I'm curious if there was any mass closure effort comparing the total mass predicted using the approach of Fig. 5 with the OPC measurements? What's the basis of the statement in L640? It's better to be quantitative in L640 instead of most include a percentage.
15. L568: does it no make sense to remove the festival period in the PMF to see how different results will be without the influence of such an event?
16. L595: can you please elaborate about what common tracers of regional and traffic factors are with HOA? Or perhaps you mean with other factors?
17. L:600-601: Instead of "practically identical" please include a quantitative measure of the similarity
18. Figure S8: what is the Industrial factor in these plots? And what two slopes are referred to in the text?
19. L641: traffic emissions are the dominant source of PM (dominant primary seems to be repetitive and confusing)

Minor edits:

1. L299: typo: change to Xact
2. L337-338: type of biomass burned and regional variation in vegetation are relaying the same information so one can be deleted
3. L471: road salting?
4. L515: add a reference
5. Fig S5: I don't recommend autoscaling all the vertical axes. It's best to show the extremes separately and zoom in so actual features are shown well.
6. Fig 5 caption and L565: Brake wear
7. L575: change "transportation" to "transport"

8. L598-599: the sentence doesn't read correctly and needs to be rephrased

---

## Author Comment (AC1)

**Referee #1**

The paper by Manousakas et al. uses aerosol composition data obtained online at high-time resolution with an ACSM, AE33, and Xact to determine sources of aerosols in Athens during a month- long study. The source apportionment (SA) approach is carried out with SoFi offline and in real-time to demonstrate performance of the real-time (RT) SoFi software when combining data from these 3 instruments. Based on the statement in the abstract, this is the first study investigating combination of all these data in SoFi RT. I think the results and approach are interesting and show promise for using SoFi RT with data from ACSM and Xact. I have some concerns and would like the authors to clarify a few details before accepting the paper.

We thank the reviewer for the overall positive assessment of our work and for the thorough review and comments, which helped us improve the quality and clarity of the paper. Please find our individual responses to all comments below, in light blue.

Major comments:

1. Although from the abstract I got the impression that SA was applied on all the data together, it seems given the different levels of uncertainties and variable magnitude of species concentrations from each measurement =, the data are actually treated separately, by running two independent PMFs in one ME-2 experiment. So is that really combining data to do source apportionment? I understand that because of the reasons explained in the paper, it may not make sense to include secondary aerosol species in the PMF run of primary metal components, but then I think the language in the abstract/body of the paper/conclusion needs to be changed.

Thank you very much for this comment, it highlights an important discussion point that we agree is valuable to include in the open review process. The goal of this study was not to merge the data into a single dataset for a source apportionment analysis, but rather to utilize the data from the three different instruments in a robust and complementary way, in order to extract as much information as possible about PM sources using a reliable methodology.

In addition to the challenges discussed in the manuscript, such as the different data types and non-equivalent uncertainties, a key reason behind our approach was to propose a methodology that is not only scientifically sound but also applicable in near real-time settings. Our aim is to support not only scientists but also environmental agencies and monitoring networks.

Unlike offline analyses performed by experts, near real-time applications require methods that are inherently robust, as there is little or no opportunity for manual correction or post-analysis adjustment. The only way to ensure usability under such conditions is to simplify the procedure while maintaining scientific rigor.

We tried to be careful with our phrasing to avoid confusing readers and aimed to clearly describe how SoFi RT handles the data. However, based on your comment, we have further revised the manuscript with particular attention to language clarity and have introduced the analytical approach earlier in the text to enhance understanding.

The manuscript was revised in the following sections (line number refer to the version of the manuscript with tracked changes):

Line 19: To avoid using the word "combined" even if it not refers to the data from the tree instruments we changed the "(AXA) set up combined with SoFi RT" to "(AXA) set up integrated with SoFi RT".

Lines 22 - 26: We added a detailed description on how the data are treated/utilized in the abstract: "SoFi RT handles data from the AXA instruments as separate inputs within a single matrix, placing them in distinct diagonal blocks. Each main instrument's data (ACSM, Xact) is processed independently, with the model applying instrument-specific constraints and generating separate source factors, effectively performing two parallel source apportionments in a single ME-2 run. Equivalent sources identified across the two instruments are then combined post-analysis to provide a unified interpretation of source contributions. The apportionment of BC to $BC_{sf}$ and $BC_{lf}$ can be performed in either of the main instrument experiments and does not require dedicated processing"

Line 50: "a combination of instruments" was substituted by "an ensemble of instruments"

Line 115: "combined with SoFi RT in Athens" to "integrated with SoFi RT in Athens"

Lines 297 – 299: The ""First, the data from both instruments are automatically combined into two diagonal blocks of a single input matrix." was revised to ""First, the data from both instruments are automatically arranged into two diagonal blocks of a single input matrix."

Line 308 - 310: The "As discussed, the data of the instruments are not combined prior to the analysis." was revised to "As discussed, the data from each instrument are treated independently in the analysis."

Line 564: The "When using the combination of AXA instruments in SoFi RT…" was revised to "When operating the AXA suite of instruments in SoFi RT…"

Line 679 – 680: The "A significant methodological insight from this work is the approach to combining data from the AXA setup." was revised to "A key methodological contribution of this work is how data from the AXA setup are utilized in parallel."

I think the paper also needs to better explain what is the advantage of running both PMFs in one ME-2 run rather than separately (other than more manual work at the end to combine the results if ran separately).

We thank the reviewer for the comment. As also discussed in the previous response, we hope that this methodology will support not only scientists but also environmental agencies and monitoring networks. To effectively achieve this, the workload and complexity of implementing the approach must be minimized.

Using a single ME-2 run offers several (primarily practical) advantages:

a) Two separate runs would either need to be performed consecutively or via two separate Igor experiments running in parallel.

b) If the experiments are run in parallel, two ME-2 engines are required, as only one can run at a time per instance.

c) Running two experiments and two ME-2 engines in parallel significantly increases the risk of crashing the model, which could interrupt online operation.

d) Two experiments increase the computational load on the PC, which may also be used to operate instruments within a monitoring network.

e) The combination and investigation of the results would have to be performed manually by the user.

All these practical considerations, along with the fact that ease of implementation is likely to be a key factor in whether such techniques are adopted by non-specialists, make this approach highly desirable.

A section was added to the manuscript to highlight these implementation advantages:

Lines 320 – 322: The following was added in the text: "This approach offers several practical advantages, including reduced computational demand, minimized risk of model crashes during real-time operation, and simplified implementation without the need for multiple ME-2 engines for separate analyses but most importantly avoid any manual post-combination of PMF results. These features make the method particularly suitable for use by non-specialists in routine monitoring environments."

2. This is a more philosophical comment, but I'd like to hear the authors thoughts on this. As shown in Fig. 5, secondary OA, sulfate, primary HOA, BC, BBOA are a large fraction of PM1. Of course this is a picture for Athens only, but I think in most other urban areas this is still the case. If I understood the conclusions correctly, gaining a valid representation of factors that can have variable profiles depending on time of year or location by RT SoFi requires some knowledge about their actual profile

before running RT SoFi. So what is the advantage of RT SoFi if one needs to also run SoFi offline to gain that initial knowledge and how often should the offline SoFi be done if we think profiles of some factors changes with time (say changes in fleet, fuels, etc)?

We thank the reviewer for the question, which is far from philosophical and instead touches on a critical aspect of operating the real-time model. Indeed, initiating the RT model requires some prior knowledge of the expected sources in the region, including their number and profiles. This information can be obtained through various means, such as using data from previous studies in the same region, extrapolating from studies in similar areas, or conducting a short exploratory offline study before implementing the RT model.

The latter approach involves using SoFi in offline mode with a small subset of regional data to define the model's starting parameters before switching to real-time operation. The more accurately these initial parameters reflect the local conditions, the better the RT model will perform.

While conducting an exploratory study in the implementation region may be the most reliable way to set up the model, it is not the only viable option. As described in Section 3.4 of the manuscript, we also tested alternative approaches to configure the model without prior detailed knowledge. The results show that using more generic initial parameters is possible, although it reduces accuracy for some sources.

Regardless of the method used to set up the initial parameters, whether via an exploratory study or generic assumptions, the benefits of the RT model remain substantial. The exploratory study can be based on historical data or require only a limited dataset (e.g., one month of data). Once real-time operation begins, the user can obtain source contribution estimates within minutes of the measurements. In contrast, classical offline approaches typically deliver source apportionment results months or even years later, which limits their usefulness for timely policy interventions.

One potential implementation scenario involves a scientific group providing the initial model parameters to an environmental monitoring network, which can then run the RT model autonomously and obtain near-real-time source contributions without further expert intervention.

Regarding changes in chemical composition over time, several approaches offer flexibility for the model to adapt to such variations. One method involves adjusting how tightly the model is allowed to deviate from the anchor profiles by setting an appropriate a-value. Another is the rolling-window approach, which enables the model to capture temporal changes by applying the analysis to shorter time intervals. Additionally, the use of partial constraints, providing only partial information about the chemical composition of a source, reduces the rigidity of the model and allows more adaptive behavior.

We have also tested new strategies, such as the "zero/nonzero" approach described in Section 3.4, which further enhances the model's ability to adjust source profiles over time. While all of these techniques increase the model's adaptability, this methodology is still new. Therefore, extended periods of real-world data are needed to fully evaluate their long-term performance.

That said, because source composition changes are typically gradual, it is reasonable to assume that the real-time model can perform reliably for at least a few years, depending on local conditions, while if progressive profile changes are identified, it is also possible to allow the anchor profile to vary over time based on previous averages.

Technical comments:

1. Sampling section: Although this is an analytical method paper, types of inlets used with each instrument should be included (PM1 cyclone, etc).

We thank the reviewer for pointing this out. Details regarding the inlet configurations of the instruments used have now been added to the Methodology section of the manuscript. Please also refer the reviewer to our response to Reviewer 2 concerning the size fractions associated with each instrument.

2. What was the time resolution of AE33? Also, are the data from AE33 (BClf, BCsf) just used in the last step when combining concentrations of different PM factors/species? L149 indicates PMF was used on AE33 but I don't see eBC showing up in any of the Xact or ACSM profiles.

The time resolution of AE33 was 1-min. This information was added in the manuscript under the instruments section (Line 137).

Real time source apportionment was conducted on eBC data using SoFi Pro, by providing the alpha values for segregation between solid fuel and biomass burning BC, and the results were incorporated in the final pie charts (Fig. 5). The eBC was not incorporated in the mass spectrum since separate apportionment took place for the eBC data only.

3. Why wasn't Ba used as one of the main Xact species? Ba is a good marker for brake wear.

Ba was below the detection limit, therefore it was not added in the analysis.

4. L194: what threshold for S/N of elements was considered?

The S/N threshold used was 0.5. The following was added in the manuscript:

Lines 210-211:

"The elements selected for the analysis were based on their signal-to-noise ratio (S/N) using a threshold equal to 0.5, and their…"

5.  L228: typically more information from PMF results are included in SA papers to justify the choice of specific solutions. Can they authors add for example Q/Qexp plots as a function of factor number?

We agree with the reviewer that the methodology used to determine the number and type of factors, as well as the overall uncertainty of the source apportionment (SA) approach, must be thoroughly documented. It is important to note that we applied a rigorous methodology to evaluate the uncertainty of the analysis, as described in lines 209–239. This included a large number of model runs combined with bootstrapping and perturbation analyses. The results demonstrate that the solution is highly stable, as selecting an incorrect number of factors typically results in increased rotational ambiguity and, consequently, higher uncertainty in the outcome.

The Q/Qexp plot is shown below. In addition to this plot, that shows a reasonable ratio for the 4-factors solution (Q/Qexp=1), the residual plots and the nature of the additional factors in the higher number of factors solution also contributed to the decision of the number of factors.

[Figure]

6.  L236: can you please explain why a higher 'a-value' would affect the results for ACSM but not Xact?

ACSM factors, particularly the POAs, often share common variables, making them more difficult to separate. For this reason, tighter constraints are generally recommended. The effect of high a-values on ACSM and Xact datasets is discussed in more detail in lines 402–428. Briefly, high a-values (e.g., 1) are more suitable for Xact data because Xact profiles typically contain several zero values for specific elements. These zeros serve as strong anchors in the PMF solution, helping to maintain the distinct identity of each factor even when the model is given substantial flexibility. In contrast, ACSM profiles rarely contain zero values, increasing the risk of factor mixing or identity loss under high a-value settings. Additionally, ACSM values are more or less in the same range, whereas for Xact there exhibit much stronger numerical differences. As a result, ACSM factors are more susceptible to rotational ambiguity when looser constraints are applied.

7.  L252: although it's mentioned later in the paragraph that the reference study was for a yearlong, I think that information needs to be mentioned in the beginning of this sentence.

We thank the reviewer for the comment. The length of our previous study was introduced in the sentence.

Line 270: "…identified five organic matter (OM) factors at the Demokritos station in Athens for a yearlong dataset (2017-2018):"

8.  L330-331: what other model parameters are needed to be defined?

Some of the parameters were included:

Line 353: "…c) various other model parameters (e.g. number of iterations, a-values, window length, etc.)."

9.  L406- starting in this paragraph is where the zero/non-zero approach for constraining the profiles is discussed. It's unclear still to me what basis is used to use the 'random values'. Why are the values for the same element for the same factor not the same in the base run and RT run? I'm still very confused how this approach sets the relative concentrations but not the relative composition.

We thank the reviewer for the comment. This approach was used to define the identity of the factors, but not to fix their relative chemical composition. When PMF is run, factors do not have fixed positions across different solutions, meaning that the same source (e.g., sea salt) may appear in the first position in one run and in a different position in another. This variation complicates averaging across runs, as mismatched factors may be combined, resulting in nonsensical profiles. To prevent this and ensure consistent factor ordering, as well as to support factor separation, constraints can play an important role.

However, as previously discussed, applying constraints reduces the model's flexibility to adapt to temporal changes in source composition. Ideally, we aim to

constrain only the minimum amount of information necessary to help the model identify the factors, while allowing the chemical composition to be determined directly from the data. This balance improves both accuracy and adaptability.

Zero values in the profile matrix have a strong impact on the PMF solution. They significantly reduce rotational ambiguity and help fix factor positions. When many variables are set to zero for each factor, and there's minimal overlap in zeroed variables between different factors, these zeros effectively anchor the factors, maintaining their identity and distinctiveness.

Building on this principle, we developed a novel approach to creating constrained profiles. This method involves setting specific variables to zero, those considered irrelevant for a given factor, based on prior source apportionment results, while assigning random initial values to all other variables. This means that no prior assumptions are made about the relative concentrations or chemical compositions, allowing the model to estimate them freely. In practice, this translates to selectively constraining only certain variables to zero, while leaving the rest of the profile entirely unconstrained.

The results of the base run are derived from a fully unconstrained model run, whereas the RT runs in this case are based on a partially constrained model run, in which only specific elements are set to zero while all other elements remain unconstrained.

10. What interpreting the Regional sources, I think a better use of WD and location of sources can be helpful. Can a map and wind roses be drawn. Is there a specific direction that has the heavy oil/industrial sources vs. port emissions?

We fully agree with the reviewer that wind roses and trajectory-based statistical methods are reliable tools for identifying regional sources. However, we chose not to use them in this study because the primary goal was to introduce and demonstrate the methodology, rather than to conduct a detailed source characterization for the region. Additionally, reliable meteorological data for the relevant time period were not available, which further limited our ability to apply such techniques.

11. L526: during the Greek Festival, COA, HOA and BBOA all increased. Does it mean that HOA is also not separated well from BBOA/COA?

During extreme events when one of these factors is increased, it is common to have mixing of these sources, since the model has to apportion very high concentrations.

12. Fig 4: what is the traffic factor? Is the Brake Wear? If so, rename it for clarity

We have revised Fig 4 accordingly.

13. Fig. 5: I find this pie chart a bit confusing/misleading. NO3- and SOA and maybe even NH4+ all likely have contributions from traffic-related emissions (NOx, VOC, and NH3 emissions). Perhaps a better categorization would be "primary traffic" instead of "traffic". Biomass should be "Biomass burning". These clarifications still don't fix the issue of mixing sources with composition. I think the best is to not mix these two concepts and actually have individual pie segments for each composition category (i.e., a segment for HOA, brake wear, etc.).

We have revised Fig. 5 accordingly. By providing both Figures 4 and 5, we aim to showcase the different graphical representation options that the model offers for displaying source contributions. Users can choose whether they prefer to present the sources as individual segments (Fig. 4) or as the sum of equivalent sources (Fig. 5), depending on the intended analysis or communication needs.

14. Fig 5: I'm curious if there was any mass closure effort comparing the total mass predicted using the approach of Fig. 5 with the OPC measurements? **What's the basis of the statement in L640? It's better to be quantitative in L640 instead of most include a percentage.**

We indeed performed a mass closure of the sum of sources and inorganic ions from Fig. 5 with the $PM_{2.5}$ fraction from the OPC and found that the AXA sum was equal to 1.01 of the $PM_{2.5}$ mass, with $R^2 = 0.73$. The part was also revised to include a percentage.

15. L568: does it no make sense to remove the festival period in the PMF to see how different results will be without the influence of such an event?

This specific event was removed in an effort to understand whether the solution would be affected during our initial assessment of offline PMF, but no change was observed, which makes sense since this festival only lasts for a few hours one day.

16. L595: can you please elaborate about what common tracers of regional and traffic factors are with HOA? Or perhaps you mean with other factors?

Yes, we mean that these factors have common tracers with other factors (e.g. HOA and COA).

17. L:600-601: Instead of "practically identical" please include a quantitative measure of the similarity

We agree with the reviewer. The sentence was revised to "within 20% difference in most cases" (Line 630).

18. Figure S8: what is the Industrial factor in these plots? And what two slopes are referred to in the text?

We thank the reviewer for identifying this typo. Figure S8 has been revised. The two slopes refer to Tyre wear source contributions, and appear when not using a tailored source profile, and the RT cannot identify the factor effectively in all cases.

19. L641: traffic emissions are the dominant source of PM (dominant primary seems to be repetitive and confusing)

The sentence was modified to:

Line 670: "The analysis showed that traffic emissions are the dominant primary source of PM…"

Minor edits:

We thank the reviewer for noticing these points that needed editing, they were taken care of.

1. L299: typo: change to Xact

2. L337-338: type of biomass burned and regional variation in vegetation are relaying the same information so one can be deleted

3. L471: road salting?

4. L515: add a reference

5. Fig S5: I don't recommend autoscaling all the vertical axes. It's best to show the extremes separately and zoom in so actual features are shown well.

6. Fig 5 caption and L565: Brake wear

7. L575: change "transportation" to "transport"

8. L598-599: the sentence doesn't read correctly and needs to be rephrased

The sentence was rephrased as follows:

Line 627: "For these reasons, the factor is more sensitive to the BS runs and the stability of the results is affected by the windows that include the high intensity events"

---

## Author Comment (AC2)

**Referee #2**

The paper introduces a novel methodology for identifying the sources of both inorganic and organic components of particulate matter. By integrating non-refractory PM compounds with elemental and black carbon data, the authors evaluate model performance using optimal initial parameters compared to results obtained with minimal prior knowledge. This approach is innovative and merits publication in *Egusphere*. The study aligns well with the journal's scope and is generally well presented. Below, I offer several minor suggestions that I believe could enhance the clarity of the paper:

We thank the reviewer for the positive feedback. Please find the responses to the individual comments below.

- **Line 150:** At this point in the manuscript, the definition of "generic conditions" is somewhat unclear. Although the meaning becomes clearer later in the text, providing a brief clarification earlier would greatly improve readability.

The sentence was modified as follows:

Line 162-166: "With respect to the implementation of the RT, the model's performance was evaluated under two conditions: using optimal initial parameters, derived from site-specific information concerning the number and chemical composition of the sources, and under less optimized, generic conditions, which relied on non-site-specific data informed by general knowledge of source composition."

- **Sofi Model:** Line 70 mentions that in the previous publication (Chen, Canonaco, Slowik et al., 2022a), where ACSM data were analyzed, an earlier version of Sofi-RT was used. While section 2.3.1 describes the model's operation in detail, it would be helpful to state early in the manuscript whether the version applied here differs from the earlier one used for evaluating ACSM and Aethalometer data.

We thank the reviewer for the comment. We revised the lines 76 – 80 to:

"Although this study employed state-of-the-art optimized SA approaches, it focused exclusively on the organic fraction of PM rather than the total PM mass, and it did not utilize the capabilities of the latest software version to process data from the Xact instrument."

The SoFi version used in our study was integrated in the manuscript:

Line 187: "SoFi RT v.9 (Source Finder Real-Time) by Datalystica was the tool used…"

- **Line 77:** I believe there is an inconsistency in this paragraph, which states: "In this study, data were collected from an Xact, an Aethalometer, a total carbon analyzer, and low-cost sensors." To my understanding, data from neither a total carbon

analyzer nor low-cost sensors were actually used. However, for the Sea Salt/Dust contribution assessment, measurements from a Grimm instrument were included (see Figure S4), yet this device is not mentioned here. Please clarify or correct this.

The authors were referring to the study by Prakash et al. (2021). This was rephrased to be clearer:

Line 83-85: "In another study that took place in Delhi, India, an RT-SA methodology that reports the results online has been set up (Prakash et al., 2021). Data were collected from an Xact, an Aethalometer, a total carbon analyzer, and low-cost sensors. Due to the nature of the input data in Prakash et al. (2021), the source apportionment focused primarily on the speciation of elements, with no information provided about secondary species."

- **Lines 127-132:** The methodology for handling Aethalometer data lacks clarity. While prior publications by the authors detail their approach to this dataset, the current paper does not fully explain how the multiple scattering effect is accounted for. Please clarify the compensation method for this effect and specify the inlet size used (e.g., PM2.5).

We thank the reviewer for the comment. The methodology for processing Aethalometer data we follow is based on the procedures established by the EBAS database. The AE33 instrument, utilizing its dual-spot technology, automatically corrects for the filter loading effect. The aerosol absorption coefficients at all wavelengths are calculated using the equivalent black carbon (EBC) concentrations, the instrument's specific sigma values, and the H-correction factor provided by EBAS. These absorption coefficients are then adjusted to standard conditions (0 °C, 1 atm).

To obtain an EBC value that more closely corresponds to Elemental Carbon (EC) as measured by EC/OC instruments, a site-appropriate Mass Absorption Cross-section (MAC) value is applied. This is the standard procedure we follow when processing such data.

However, in this study, the AE33 was feeding the data into the RT software in near real-time mode. As a result, the raw EBC data provided by the instrument were directly used by the analysis software without post-processing. To avoid confusion, we have updated the manuscript to reflect this approach, as follows:

Line 137-140: "In this study, the AE33 operated with a $PM_{2.5}$ inlet, and minute-resolved EBC mass concentrations at the relevant wavelengths were used as raw data from the instrument under near real-time conditions."

- **Line 235:** The authors assert that high a-values are unsuitable for ACSM data. Include a reference or provide justification for this claim.

We agree with the reviewer that this statement deserves further justification. On previous studies conducted worldwide it has been found that for the ACSM data if we

constraint with a high a-value the chemical composition of the factor can change significantly to the point that their identification is no longer clear. For that reason it is suggested that the upper threshold is 0.5 for more variable sources such as BBOA, while for the factors that refers to primary organic aerosols the suggested value is usually even lower ~0.2. These suggestion are described both in earlier publications as well as in the publication that refers to the harmonized source apportionment analysis protocol in Europe (Crippa et al., 2014) (Chen et al., 2022).

A reference that suggests tight constraints on the primary factors of the ACSM was included in the text:

Line 252: "Higher a-values are not suitable for ACSM data, as the identity of the factors can change (Crippa et al., 2014; Chen et al., 2022)., in contrast to Xact data."

- **Line 407:** The construction of the constraint profiles remains unclear, particularly the criteria for identifying variables that can be classified as "irrelevant." Please elaborate on this process.

We thank the reviewer for the comment. We define 'irrelevant variables' as those that are not identified in previous studies as being emitted by specific sources. For example, silicon (Si) is not typically associated with brake wear; therefore, its contribution to this factor was set to zero. The determination of relevant variables for factor identification was based on established knowledge of source compositions, findings from prior studies, and the optimized source apportionment results obtained in the present study. The following statement has been added for clarification:

Line 320-322: "This approach involved setting certain variables to zero (those deemed irrelevant for factor identification, on established knowledge of source compositions, findings from prior studies, and the optimized source apportionment results obtained in the present study)"

- Although the authors clearly state that ACSM analyzes PM1 while Xact measures PM2.5, the source apportionment percentages in Figures 4 and 5 are reported without distinction between these size fractions. Additionally, the abstract refers generically to contributions to "PM mass" without specifying size ranges. Given that each source inherently exhibits its own particle size distribution, we recommend elaborating on this concept and explicitly addressing the limitations of presenting combined results across different size fractions.

We thank the reviewer for bringing up this important point. Based also on the suggestion of Reviewer 1, we have now specified the size fraction measured by each instrument used in this study.

Although we acknowledge that using consistent size fractions across all instruments is preferable, we consider our approach to be justified and methodologically sound for two main reasons:

(a) The data from the different instruments were not combined prior to the source apportionment analysis. Instead, each dataset was treated independently. Consequently, differences in the size distribution of sources do not directly influence the analysis, as each dataset reflects the characteristics of its respective size fraction.

(b) Carbonaceous aerosols are typically concentrated in the fine fraction. BC is primarily associated with fine particulate matter due to its formation from combustion processes (Saarikoski et al., 2021). Supporting this for the organic fraction, studies employing collocated ACSMs equipped with both $PM_1$ and $PM_{2.5}$ lenses have demonstrated that organic aerosol components in both size fractions are comparable in terms of chemical composition and mass.

For instance, Liu et al. (2024) evaluated the performance of the Time-of-Flight Aerosol Chemical Speciation Monitor with a capture vaporizer (TOF-ACSM-CV) during the RITA-2021 field campaign. Two identically configured TOF-ACSM-CV instruments were deployed to simultaneously measure non-refractory $PM_1$ and $PM_{2.5}$ at the Cabauw Experimental Site for Atmospheric Research (CESAR) in the Netherlands. The study found that $PM_1$ accounted for approximately 85% of the $PM_{2.5}$ organic aerosol mass, with both fractions exhibiting similar chemical compositions.

Similarly, in a study conducted in urban Nanjing, China, Zhang et al. (2017) deployed a $PM_{2.5}$-capable ACSM alongside a standard $PM_1$ ACSM. The measurements revealed that the mass spectra and time series of primary and secondary organic aerosols (POA and SOA) were highly consistent between the $PM_1$ and $PM_{2.5}$ size fractions, indicating that organic aerosols are similarly distributed in both.

Nevertheless, we recognize that this approach has certain limitations. The degree of similarity between fractions of carbonaceous aerosol content can vary depending on local sources, atmospheric conditions, and site characteristics. Therefore, while the findings presented here support the validity of our approach for this specific campaign, we caution against generalizing this method without site-specific validation. To reflect this, we have included the following statement in the manuscript to acknowledge the potential limitations of using mixed size fractions across instruments:

Line 149-152: " In this study, different PM fractions were used to implement the RT-SA approach. For this reason, we do not attribute the sources collectively to specific size fractions, as doing so would introduce uncertainty. It is important to note that, although the comparison of carbonaceous aerosol fractions appeared consistent in this study leading to good reconstruction of $PM_{2.5}$ mass, this may not be the case in all environments. Variations in local emission sources, atmospheric processing, and particle size distributions can lead to inconsistencies between size fractions. Therefore, where possible, harmonized size cuts should be applied, and caution should be exercised when interpreting what the total mass represents."

Liu, X., Henzing, B., Hensen, A., Mulder, J., Yao, P., van Dinther, D., van Bronckhorst, J., Huang, R., and Dusek, U.: Measurement report: Evaluation of the TOF-ACSM-CV for PM1.0 and PM2.5 measurements during the RITA-2021 field campaign, Atmos. Chem. Phys., 24, 3405–3424, https://doi.org/10.5194/acp-24-3405-2024, 2024.

Saarikoski, S., Niemi, J. V., Aurela, M., Pirjola, L., Kousa, A., Rönkkö, T., and Timonen, H.: Sources of black carbon at residential and traffic environments obtained by two source apportionment methods, Atmos. Chem. Phys., 21, 14851–14869, https://doi.org/10.5194/acp-21-14851-2021, 2021.

Zhang, Y., Tang, L., Croteau, P., Worsnop, D., et al.: Field characterization of the PM2.5 Aerosol Chemical Speciation Monitor: insights into the composition, sources, and processes of fine particles in Eastern China, Atmos. Chem. Phys., 17, 14501–14517, https://doi.org/10.5194/acp-17-14501-2017, 2017.